# Modular Pretraining Enables Access Control

**Ethan Roland** [* 1]  **Murat Cubuktepe** [* 1]  **Erick Martinez** [* 1]  **Stijn Servaes** [1]  **Keenan Pepper** [1]  **Mike Vaiana** [1]
**Diogo Schwerz de Lucena** [1]  **Judd Rosenblatt** [1]  **Addie Foote** [2]  **Cem Anil** [3]  **Alex Cloud** [3]

## Abstract

AI developers face a dual-use dilemma. An AI capability that helps one user cure a disease can help another synthesize one. This dilemma could be resolved with access control, limiting dual-use AI capabilities to trusted deployments with a legitimate need. A gold standard for access control would be to serve separate models with different capabilities to different users. However, training and deploying multiple models is prohibitively expensive. To address this challenge, we propose gradient-routed auxiliary modules (GRAM), a pretraining method that adds modules to a neural network and selectively updates them to induce specialization. Ablating a module at inference time removes its capability from the network, approximating a model trained on filtered data. We evaluate GRAM on synthetic stories and realistic dual-use data spanning virology, cybersecurity, nuclear physics, and specialized code. These experiments show that GRAM disables targeted capabilities while preserving the rest, and resists their recovery under finetuning better than post-hoc unlearning. Most importantly, a Chinchilla-optimal scaling analysis from 50M to 5B parameters shows that the gap between data-filtered and full-data models widens with scale on removed capabilities but stays small on retained ones, and that GRAM closely tracks data filtering. GRAM's training cost is independent of the number of supported capability profiles, yielding a $5\times$ reduction over data filtering in our 5-profile setting.

## 1. Introduction

Some AI capabilities are dual-use, enabling both beneficial and harmful applications (Brundage et al., 2018). For exam-

---
[*]Equal contribution  [1]AE Studio  [2]Independent  [3]Anthropic. Correspondence to: Ethan Roland <ethan@ae.studio>.

*Proceedings of the $43^{rd}$ International Conference on Machine Learning*, Seoul, South Korea. PMLR 306, 2026. Copyright 2026 by the author(s).

ple, the knowledge required to manufacture vaccines overlaps with that needed to develop biological weapons (Sandbrink & Koblentz, 2022; Drew & Mueller-Doblies, 2017); similarly, knowledge of cybersecurity can be used to fortify computer systems or to execute attacks against them (Truong et al., 2020; Roguski, 2021). Models deployed with dual-use capabilities pose misuse risk, yet withholding these models forfeits their benefits. Without mechanisms for differentiated access, AI developers face an all-or-nothing choice: the dual-use dilemma (Miller & Selgelid, 2007).

*Access control*, the restriction of access to resources based on user authorization, could help address this dilemma (Sandhu & Samarati, 1994; Wybitul, 2025). Rather than exposing every capability in every deployment, AI developers could provide model variants appropriate to specific deployments based on trust and need. This is consistent with established security principles such as least privilege (Saltzer & Schroeder, 1975; NIST, 2020), which holds that permissions should be restricted only to those required to perform a task.

For LLMs, we define a *capability* informally as the ability to perform a particular type of task, and define a *capability profile* as a collection of capabilities. A gold standard for AI access control would be to serve separately trained models with different capability profiles to different users. This standard could be achieved by data filtering across multiple training runs, each removing a different subset of the corpus to induce a specific profile. Because each resulting model never saw the filtered data, it is robust to adversarial elicitation such as finetuning (Maini et al., 2025; O'Brien et al., 2025). However, supporting $N$ capability profiles requires training and deploying $N$ separate models, which is prohibitively costly at frontier scale (Cottier et al., 2024).

This cost has motivated cheaper approximations, but many cannot robustly remove capabilities. Standard methods such as refusal training, output filtering, and classifiers restrict model outputs, but leave the underlying capability intact and are vulnerable to adversarial elicitation (Wei et al., 2023; Zou et al., 2023; Sharma et al., 2025). Post-hoc unlearning applies targeted gradient updates to remove capabilities from a trained model (Li et al., 2024; Yuan et al., 2025), but finetuning on a few examples can rapidly recover them

(Deeb & Roger, 2024; Łucki et al., 2025).

An alternative approach is model branching: train a base model on filtered data, then split into separate variants by finetuning each on a different dual-use capability. Model branching reduces pre-training cost but still requires multiple deployments, which is inefficient when some models see sparse usage. Parameter-efficient finetuning such as LoRA (Hu et al., 2022) can reduce deployment cost by sharing one base model across many adapters (Sheng et al., 2024), but may lack the capacity to learn capabilities at pre-training scale (Biderman et al., 2024; Schulman & Lab, 2025).

We propose *gradient-routed auxiliary modules* (GRAM), a pre-training method that yields multiple capability profiles in a single training run. At initialization, GRAM augments the MLP layers of a dense transformer with several smaller MLP modules. During training, GRAM selectively enables forward and backward passes for different modules based on the data in the current training batch, directing capability-specific updates to corresponding modules. At inference time, ablating a module removes its capability, and we find that the ablated model resists malicious finetuning on the removed domain about as well as data filtering, and substantially better than post-hoc unlearning.

GRAM builds on prior work, combining domain-specific modules (Gururangan et al., 2022), data-dependent selective gradient updates (Cloud et al., 2024), and separate optimizers for parameter subsets (Shilov et al., 2025). We show that GRAM extends the Pareto frontier of capability shaping, evaluating it on synthetic and realistic dual-use datasets, scaling from 50M to 5B parameters, and comparing against data filtering, model branching, and post-hoc unlearning.

**GRAM approximates separately trained, data-filtered models in a single run.** In Simple Stories, a synthetic dataset partitioned into core and auxiliary subsets, GRAM closely matches data filtering on both retention and removal.

**GRAM isolates realistic dual-use capabilities.** We train an 800M-parameter model on a core dataset of web text (Penedo et al., 2024), code (BigCode, 2023), and arXiv papers, plus four auxiliary dual-use domains: virology, cybersecurity, nuclear physics, and Lisp code. On core data, GRAM performs nearly as well as the all-data baseline model. On auxiliary capability retention and removal, GRAM matches both data filtering and a parameter-efficient branched finetuning (FT-LoRA). Removal of targeted capabilities via GRAM is more robust to malicious finetuning than when removed via post-hoc unlearning. Enabling multiple GRAM modules does not degrade performance, supporting arbitrary capability subsets from a single model. When training with partially labeled data, GRAM achieves far better capability removal than both filtering and branched

finetuning.

**Capability isolation improves with scale.** Across compute-optimal models (Hoffmann et al., 2022) ranging from 50M to 5B parameters, GRAM closely matches the capability profile of data filtering. Like data filtering, GRAM's suppression of the forget capability widens with scale, suggesting that its advantages persist at the model scaling frontier.

## 2. Background

**Problem setting.** Let $\mathcal{D} = \{\mathcal{D}_1, \ldots, \mathcal{D}_N\}$ be a set of datasets, where $\mathcal{D}_1$ is a general-purpose *core* dataset and each $\mathcal{D}_{i>1}$ corresponds to an auxiliary capability. Each dataset is partitioned into training and validation sets. Let $\ell(\mathcal{M}, \mathcal{D}_i)$ denote the validation cross-entropy loss of model $\mathcal{M}$ on dataset $\mathcal{D}_i$.

A *capability profile* $S \subseteq \{1, \ldots, N\}$ specifies which capabilities a model should have. We assume $1 \in S$ for all profiles. For a given profile $S$, we define the *retain set* $\mathcal{R} = S \setminus \{1\}$ as the auxiliary capabilities to preserve, and the *forget set* $\mathcal{F} = \{2, \ldots, N\} \setminus S$ as the auxiliary capabilities to restrict. Let $\mathcal{S}$ refer to the set of all profiles.

**Problem statement.** For each capability profile $S \in \mathcal{S}$, our goal is to produce a model $\mathcal{M}_S$ that achieves low validation loss on $\mathcal{D}_i$ for $i \in S$ (the core and the retained capabilities) and high validation loss on $\mathcal{D}_j$ for $j \in \mathcal{F}$. The total compute budget to produce all $\mathcal{M}_S$ variants is fixed to that of a single baseline training run over all data.

**Compute ratio metric.** Incremental reductions in cross-entropy loss require increasing amounts of training compute. To enable comparisons across datasets and model sizes, we report performance in terms of the compute required to reach a particular loss, relative to a baseline run. Let $\mathcal{M}_{\text{BL}}$ denote a standard Transformer with a dense MLP block, trained on the full dataset $\mathcal{D}_{all} = \bigcup_{i=1}^{N} \mathcal{D}_i$. For each dataset $\mathcal{D}_i$, we fit a monotone power-law learning curve $L_i(s)$ that maps baseline training step $s$ to the baseline's validation cross-entropy loss at that step. We denote the inverse of this curve by $L_i^{-1}(\ell)$. For any model variant $\mathcal{M}$ and dataset $\mathcal{D}_i$, we define compute ratio as

$$\text{CR}(\mathcal{M}, \mathcal{D}_i) = \frac{L_i^{-1}\big(\ell(\mathcal{M}, \mathcal{D}_i)\big)}{L_i^{-1}\big(\ell(\mathcal{M}_{\text{BL}}, \mathcal{D}_i)\big)}.$$

A compute ratio of $r$ indicates that the model achieves a loss equivalent to the baseline after a fraction $r$ of the baseline training compute. An ideal model would have a compute ratio of at least 1 for all datasets in $S$ and a compute ratio of 0 for all datasets in $\mathcal{F}$. We provide a more detailed description of the compute ratio metric in Appendix M.

**Evaluation metrics.** We evaluate models using Core ($\text{CR}_\mathcal{C}$), Retain ($\text{CR}_\mathcal{R}$), and Forget ($\text{CR}_\mathcal{F}$) performance, defined as

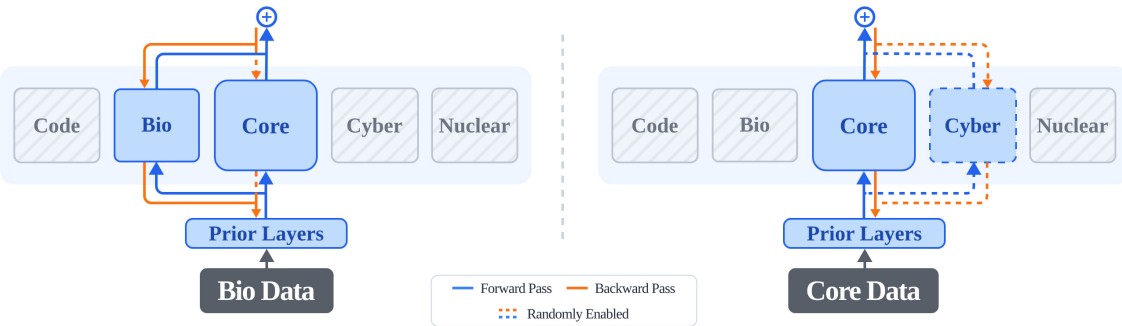

*Figure 1.* **Overview of GRAM training.** In a decoder-only Transformer, GRAM modifies each MLP block ("Core") by introducing additional auxiliary modules, one per dual-use capability. **Left:** On auxiliary data, the core MLP and corresponding auxiliary module are active in the forward pass. Gradients always update the auxiliary module and update the non-auxiliary parameters with probability $p_{as}$, enabling tunable capability isolation. **Right:** On core data, gradients always update the non-auxiliary parameters. With probability $p_{cr}$, a random auxiliary module is also activated and updated, training the core performance to be robust to auxiliary module activations. Modules not selected for a batch receive no forward activation nor gradient updates, inducing capability isolation and enabling capability removal by module ablation at inference time.

follows:

$$\text{CR}_{\mathcal{C}} = \text{CR}(\mathcal{M}, \mathcal{D}_1);$$

$$\text{CR}_{\mathcal{R}} = \frac{1}{|\mathcal{R}|} \sum_{i \in \mathcal{R}} \text{CR}(\mathcal{M}, \mathcal{D}_i);$$

$$\text{CR}_{\mathcal{F}} = \frac{1}{|\mathcal{F}|} \sum_{i \in \mathcal{F}} \text{CR}(\mathcal{M}, \mathcal{D}_i).$$

Higher compute ratios (and lower loss values) indicate better core and retain performance. Lower compute ratios (and higher loss values) indicate better forget performance.

*Elicited Forget Performance* measures whether excluded capabilities are truly removed rather than merely suppressed (Hofstätter et al., 2025; Lee et al., 2025; Donoway et al., 2025). We finetune each model on a fixed sample from each forget category (512 sequences, approximately 0.5M tokens, in our dual-use and scaling experiments; see Appendix B) and select the checkpoint achieving the lowest validation loss. Sample size is held constant across model sizes and methods. Elicited forget is then calculated identically to $\text{CR}_{\mathcal{F}}$. Lower compute ratios after finetuning indicate more robust removal. This budget models a limited adversary: with substantially more elicitation data or compute, partial recovery should be expected even from data-filtered models (O'Brien et al., 2025; Deeb & Roger, 2024).

## 3. Gradient-Routed Auxiliary Modules

GRAM allows a single model to have multiple capability profiles. It achieves this by controlling which modules receive gradient updates based on the data label. Because each module is trained on a specific type of data, removing a module at inference time eliminates the corresponding capability while preserving the others.

**Architecture.** GRAM builds on a standard Transformer architecture by augmenting each MLP block with smaller, auxiliary MLP modules. Unlike a mixture-of-experts architecture (Shazeer et al., 2017), GRAM has no learned router and no per-token routing; instead, gradient routing directs updates to specific modules based on data labels, as illustrated in Figure 1. We find that alternative architectural choices can yield equivalent results under fixed parameter counts. Further details are provided in Appendix E.

Each GRAM layer contains one large *core MLP* and $N-1$ *auxiliary modules*, corresponding to the auxiliary datasets $\mathcal{D}_2, \ldots, \mathcal{D}_N$. Unless otherwise noted, the core module is the same size as the baseline Transformer's MLP layer. Each auxiliary module has a small fraction of the parameters of the core module. The core module captures general knowledge, whereas auxiliary modules capture capability-specific knowledge.

Module activation is defined by a binary indicator $m \in \{0,1\}^N$, where $m_1 = 1$ always (the core MLP is active on every forward pass) and $m_{i>1}$ selects auxiliary modules. The GRAM layer output is then

$$h_{\text{out}} = \sum_{i=1}^{N} m_i \, \text{E}_i(h_{\text{in}})$$

where $h_{\text{in}}$ is the input to the GRAM layer and $\text{E}_i$ is the $i$-th module. We detail how the module participation in the forward and backward passes is controlled in the following paragraphs.

**Gradient routing.** GRAM achieves capability isolation via *gradient routing*, with dataset-dependent forward pass activations and gradient updates over disjoint parameter subsets. We partition the model parameters into one core set (all shared Transformer parameters, including embeddings, attention, normalization layers, unembedding, and the core

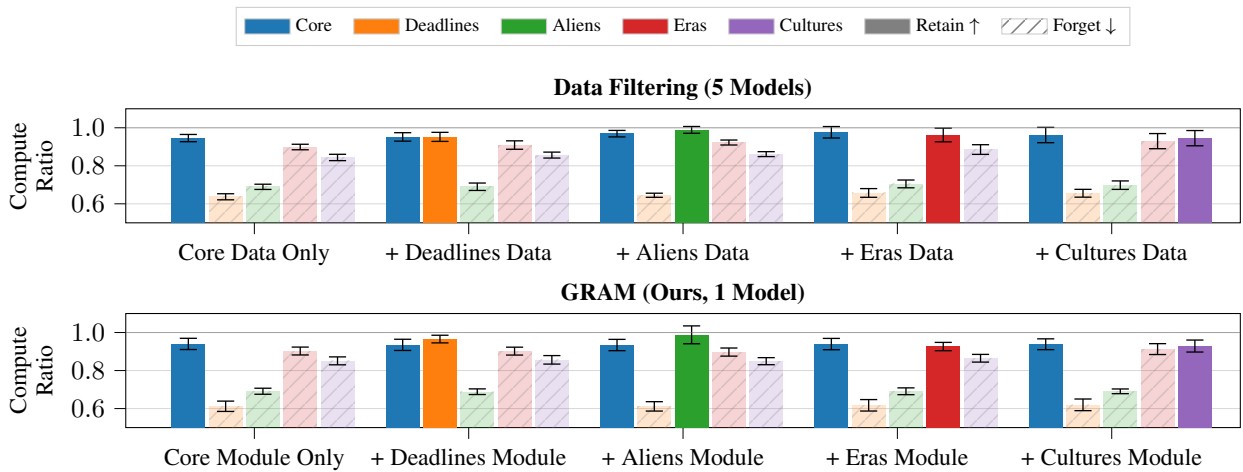

*Figure 2.* **GRAM approximates five separate models trained with data filtering**. Each group of bars corresponds to a model or model configuration, and each bar within a group shows the performance, in terms of compute ratio, on a different evaluation dataset. The top row shows data filtering, where each model is trained on the core dataset plus at most one auxiliary dataset. The bottom row shows GRAM, where the core MLP is always active and at most one auxiliary module is enabled. Solid bars show performance on core and retain datasets, where higher values indicate better retention. Hatched bars show performance on forget datasets, where lower values indicate more effective capability removal. Error bars show 90% CIs for the mean over $N = 3$ independent training runs.

MLP) and $N - 1$ disjoint auxiliary sets, each containing a single small auxiliary module. Following Shilov et al. (2025), we optimize each set of parameters using a separate AdamW optimizer (Loshchilov & Hutter, 2019).

For each training batch, GRAM specifies the data trained on, the modules active in the forward pass, and the parameter partitions updated in the backward pass. Three hyperparameters control batch composition: Auxiliary Factor $p_{\text{af}} : \{2, \ldots, N\} \to \mathbb{R}_{>0}$, Auxiliary Spread $p_{\text{as}} \in [0, 1]$, and Core Robustness $p_{\text{cr}} \in [0, 1]$.

*Core batches.* For a batch from the core dataset $\mathcal{D}_1$, the core MLP is always active in the forward pass. With probability $p_{\text{cr}}$, a single randomly sampled auxiliary module is also activated. All active parameters are updated in the backward pass. Increasing $p_{\text{cr}}$ encourages stable core performance that is robust to the presence or absence of auxiliary module activations. However, setting $p_{\text{cr}}$ too high can degrade both core and retain performance, due to interference from randomly activated auxiliary modules. Appendix F reports the empirical tradeoffs for $p_{\text{as}}$ and $p_{\text{cr}}$.

*Auxiliary batches.* For a batch from an auxiliary dataset $\mathcal{D}_{i>1}$, the forward pass activates both the core MLP and auxiliary module $i$. In the backward pass, auxiliary module $i$ is always updated, while the core parameters are updated with probability $p_{\text{as}}$. Increasing $p_{\text{as}}$ allows auxiliary gradients to update core parameters, increasing competence on the capability but reducing modularization. $p_{\text{af}}(i)$ sets the sampling frequency of dataset $\mathcal{D}_i$; when oversampling, we undersample $\mathcal{D}_1$ to maintain compute-equality with the baseline.

**Inference and capability control.** At inference time, a capability profile $S$ is served by activating the core MLP together with each auxiliary module $i \in S$ and ablating the rest.

## 4. GRAM Approximates Multiple Data-Filtered Models in a Single Run

We show that one model trained with GRAM approximates $N$ models trained with data filtering.

**Dataset and setup.** We use Simple Stories (Finke et al., 2025), a synthetic dataset of 2M children's stories, each belonging to one of 48 topic categories. We designate four categories as auxiliary capabilities (chosen alphabetically), with the remaining 44 forming the core dataset $\mathcal{D}_1$.

We train an 8-layer, 26M-parameter Transformer for one epoch (batch size 128, sequence length 256). GRAM uses an always-active core MLP ($d_{\text{core}} = 1856$) and smaller auxiliary modules ($d_{\text{aux}} = 192$), while both data filtered and baseline models use a dense hidden size of 2048. The number of active parameters per forward pass is matched across methods. We set $p_{\text{af}}(i) = 1.0$ for all auxiliary datasets $i$, $p_{\text{as}} = 0.3$, and $p_{\text{cr}} = 0.5$. All methods are trained with three random seeds (varying weight initialization, data ordering, and other stochastic operations during training) to compute confidence intervals. Detailed results, including comparisons of GRAM against methods beyond data filtering, are provided in Appendix A; qualitative sample continuations from each method in this setting are shown in Appendix K.

**Comparison methods.** We compare GRAM with data

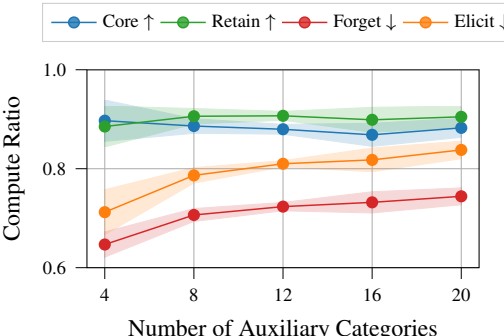

*Figure 3.* **GRAM scales to many auxiliary categories.** As the number of auxiliary categories increases from 4 to 20, retain performance remains near baseline and core stays roughly constant, while forget and elicited-forget performance remain substantially lower. Forget and elicited-forget rise modestly with more categories, but a clear separation persists across the full range. Shaded regions show 90% CIs for the mean over $N = 3$ independent training runs.

filtering, which trains separate models on $\mathcal{D}_1 \cup \mathcal{D}_i$ for each auxiliary story topic $i$, plus a core-only model trained on $\mathcal{D}_1$.

**Results.** Figure 2 shows that across five capability profiles (defined by different story categories), GRAM approximates the performance of distinct models trained with filtering.

For example, when *Alien Encounters* is retained, both filtering and GRAM nearly match the baseline (0.99 each), while a forgotten category such as *A Deadline or Time Limit* stays well below (0.65 for filtering, 0.61 for GRAM). This separation holds across all five profiles: retained capabilities stay near baseline and forgotten ones drop well below. Core performance stays near baseline throughout (GRAM 0.94, filtering 0.96), so the gap reflects targeted removal rather than a weaker shared model. Aggregated across configurations, GRAM reaches a compute ratio of 0.95 on retained auxiliary capabilities, closely matching data filtering's 0.96.

**Scaling the number of auxiliary capabilities.** We also evaluate GRAM on Simple Stories as the number of auxiliary capabilities grows, holding the total training budget and the core/auxiliary token mixture fixed at 80%/20% of the full dataset while varying the number of auxiliary categories from 4 to 20 (so the fixed auxiliary budget is split across progressively more topics). Figure 3 shows that retain performance stays near baseline across the full range, indicating that the model learns and separates many auxiliary categories at once. Forget and elicited forget performance remain well below retain, and core performance stays roughly constant (around 0.88 across the range), so the separation between retained and suppressed capabilities persists as the number of modules grows. Full details are provided in Appendix D.

## 5. GRAM Isolates Dual-Use Capabilities Learned From Real Data

We evaluate whether GRAM remains effective when capabilities reflect the complexity of realistic domains.

**Dataset and setup.** Our core dataset combines FineWeb-Edu (Penedo et al., 2024) with general papers from arXiv.org and code from The Stack (BigCode, 2023) with the Lisp programming language removed.

Our auxiliary domains consist of four datasets: virology (Europe PMC, a repository of biomedical and life-sciences literature), cybersecurity (arXiv), nuclear physics (OSTI, the U.S. Department of Energy's research repository, and arXiv), and specialized code (The Stack). For specialized code we use Lisp as a proxy for a niche or proprietary codebase, since it is rare relative to the mainstream languages in the core. Including general-purpose arXiv papers in the core, processed identically to the auxiliary papers, ensures that differences in compute ratio reflect domain-specific knowledge rather than formatting artifacts. The exact sources and arXiv categories for each domain are given in Appendix B.

We train 800M-parameter models for one epoch on 16B core tokens plus 160M auxiliary tokens (1% of the core dataset), with approximately 40M tokens per auxiliary category (Table 2). GRAM uses a core MLP with $d_{\text{core}} = 6400$ and four auxiliary modules with $d_{\text{aux}} = 640$ (49.2M parameters each) with $p_{\text{as}} = 0.5$, $p_{\text{cr}} = 0.2$, and $p_{\text{af}} = 4.0$ for code and $p_{\text{af}} = 3.0$ otherwise.

**Model branching.** In our model branching experiments, the shared base model is trained on core data only, and each branch is then finetuned on a single auxiliary dataset mixed with core data. The inclusion of core data in finetuning is sometimes called rehearsal (Scialom et al., 2022). We parametrize rehearsal with the Core-Auxiliary Ratio ($p_{\text{ca}} \in \mathbb{R}_{\geq 0}$), which is the ratio of core to auxiliary batches seen during finetuning. The hyperparameter $p_{\text{lr}}$ controls the learning rate during the finetuning phase, expressed as a proportion of the learning rate used in the base phase. Similarly to GRAM, $p_{\text{af}}(i)$ controls the frequency with which batches are drawn from $\mathcal{D}_i$, enabling over- or under-sampling of the auxiliary data.

**Comparison methods.** We compare against five methods: (1) **Baseline**, an 800M-parameter Transformer trained on all data (core and auxiliary domains), used as the reference for the Compute Ratio metric defined in Section 2, which scores exactly 1 by definition; (2) **Filtering**, which trains separate data filtered models as in the previous section; (3) **FT-LoRA** (Hu et al., 2022), a branched training baseline that finetunes capability-specific low-rank adapters with rehearsal on top of a frozen core-only model, whose core:auxiliary ratio hyperparameter we analyze in Appendix F; (4) **FT-Full**, which follows the same proce-

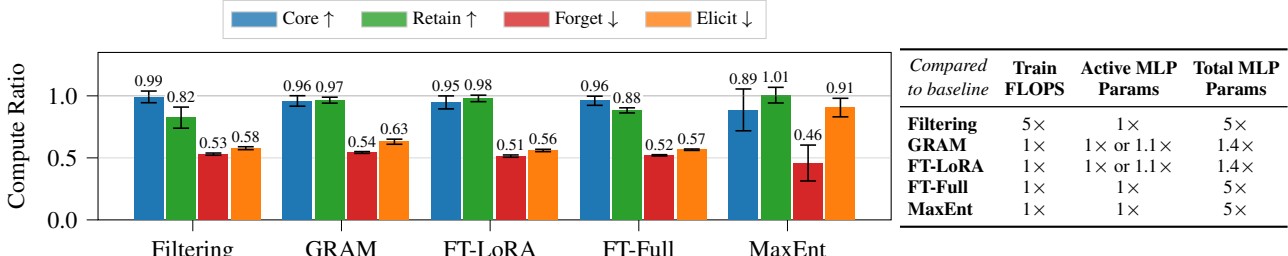

*Figure 4.* **GRAM matches data filtered performance in a realistic setting. Left:** Compute ratio per method and data class. Each bar shows the mean performance over the five capability profiles produced by that method. For example, the value of Core for filtering (0.99) is the mean compute ratio across five training runs that used data filtering with different retain sets. **Right:** Training costs and parameter counts relative to the baseline. Both FT-LoRA and GRAM closely match data filtering in performance while avoiding its per-domain training cost. FT-Full achieves similar performance but requires deploying multiple models. MaxEnt has low compute ratio on Forget but a high compute ratio on Elicit, indicating that MaxEnt does not robustly remove model capabilities. Error bars show 90% CIs for the mean over $N = 3$ independent training runs.

dure as FT-LoRA but finetunes all parameters of a separate model copy per capability instead of low-rank adapters; (5) **MaxEnt** (Yuan et al., 2025), a post-hoc unlearning method that maximizes output entropy on forget data. All methods except filtering use the same training compute. Further details are in Appendix B.

We also consider **DEMix** (Gururangan et al., 2022), which similarly uses domain-specific modules but lacks a shared core module. We evaluate DEMix in our synthetic experiments (Appendix A) but omit it from our main results, as it performs poorly under class imbalance. Without a shared core, each module must redundantly learn basic language structure from limited data.

**Training and inference time costs.** The right-hand table in Figure 4 reports training FLOPS and the number of parameters served at inference, relative to the baseline. Filtering, FT-Full, and MaxEnt require 5× more total parameters than baseline in this setting, since each capability profile needs a distinct model. In contrast, both GRAM and FT-LoRA match baseline training compute for smaller increases in active and total parameters.

**Results.** Figure 4 shows retain and forget performance alongside the compute and parameter requirements of each method. Retain and forget values are means over all $N = 5$ capability profiles (each of which defines a different retain and forget set). Filtering serves as a reference, requiring five times as many FLOPS to train. All other methods use the same FLOPS as the baseline.

GRAM and FT-LoRA approximate data filtering well, reaching roughly equal compute-normalized losses on the core, retain, and forget sets while adding at most 10% more active parameters to the MLP layer. FT-Full reaches similar performance but is less parameter efficient. MaxEnt performs worst: the degradation it induces on the forget set is mostly reversed by finetuning, as reflected by the elicited forget

metric. GRAM and FT-LoRA perform comparably here, but with small quantities of auxiliary data and a single training epoch, our setting is favorable to low-rank adaptation, and it may not be competitive in regimes with longer finetuning phases (Biderman et al., 2024). We use the remainder of this section to examine where the two methods diverge.

**Arbitrary capability subsets.** Figure 5 shows GRAM and FT-LoRA when enabling multiple GRAM modules or LoRA adapters. Successful combination of arbitrary subsets of capabilities enables $2^{N-1}$ capability profiles from a single training run, implying an exponential training efficiency advantage over data filtering. Empirically, GRAM demonstrates better composability. While GRAM's performance on retained classes remains constant when additional auxiliary modules are active, FT-LoRA generally degrades when more than one auxiliary LoRA adapter is summed. Surprisingly, GRAM achieves composability despite never training in the configuration where all modules are active.

**Partial labeling.** In all other experiments, we assume every data point carries an accurate capability label. Here, we relax this assumption by labeling only 50% of the training data. GRAM extends to this setting with a single additional rule: on unlabeled batches, all parameters (the core MLP and every auxiliary module) participate in the forward and backward passes. For filtering and FT-LoRA, unlabeled data is treated as core. Consistent with prior work on gradient routing (Cloud et al., 2024; Shilov et al., 2025), GRAM achieves significantly lower forget performance ($0.60 \pm 0.01$) than either filtering ($0.85 \pm 0.01$) or FT-LoRA ($0.85 \pm 0.01$) in the partially labeled setting, indicating superior modularization (Figure 6). We discuss a candidate mechanism in Section 8 and provide additional details in Appendix G.

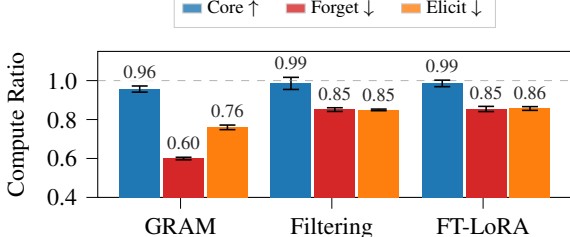

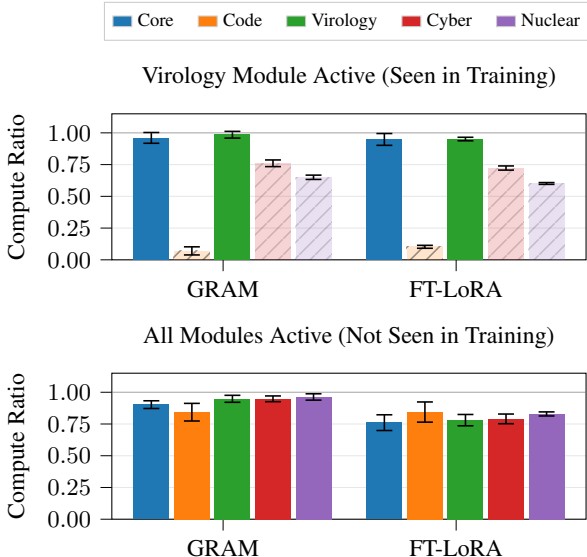

Figure 5. **GRAM improves capability composability over FT-LoRA.** Compute ratios for GRAM and FT-LoRA when only virology is retained compared to when all auxiliary capabilities are retained. GRAM has a high virology compute ratio in both cases, whereas FT-LoRA achieves worse compute ratios in all categories when all adapters are active. Error bars show 90% CIs for the mean over $N = 3$ independent training runs.

Figure 6. **GRAM achieves better modularization than alternatives under partial labeling.** With only 50% of training data labeled, GRAM reaches an aggregate forget compute ratio substantially below data filtering and FT-LoRA, while maintaining competitive core performance. Error bars show 90% CIs for the mean over $N = 3$ independent training runs.

## 6. Capability Isolation Improves with Scale

In our final experiment, we evaluate compute-optimal scaling trends for GRAM, FT-LoRA, and data filtering.

**Experimental setup.** We train models with 50M, 100M, 200M, 400M, 800M, 2B, and 5B parameters for a single epoch on subsampled datasets chosen to follow Chinchilla-optimal scaling, with auxiliary data fixed at 1% of core dataset size. For each model size, we compare: (1) **Baseline**: Standard Transformer trained on all data, (2) **GRAM**: one auxiliary module per domain, (3) **FT-LoRA**: one LoRA adapter per domain, (4) **Core + Virology DF**: a data filtering run over Core & Virology data. Both GRAM modules and auxiliary LoRA adapters have parameter counts set to 10% of the baseline model's MLP parameter count at a given scale. We report core, retain, forget, and elicitation performance for each method, restricted to results where the targeted retain label was Virology. The learning rate and batch size for each model size follow power-law scaling fits, detailed in Appendix J. We provide more details about the experimental setup in Appendix C.

**Results.** Figure 7 shows performance across model sizes for data filtering, FT-LoRA and GRAM. Data filtering improves with scale: larger models maintain consistent core and virology performance, while learning the excluded capability less well and resisting adversarial finetuning more strongly. We speculate that the decreasing compute ratio for Forget is

because, at small scale, most of the baseline's loss reduction on any domain comes from generic language structure that transfers from core data. With scaling of model and dataset size, a growing share of the baseline's improvement on the forget domain reflects domain-specific learning that the filtered model cannot acquire from core data alone, widening the gap.

Elicited forget starts high and approaches forget, which is expected because the fixed elicitation data budget represents a decreasing fraction of training data with scale, and data filtering should produce models that aren't easy to elicit from. These trends suggest data filtering scales favorably.

GRAM and FT-LoRA track data filtering closely at every scale, despite requiring five times less compute in amortized terms. GRAM shows an off-trend gain in retain performance at 5B parameters, which we attribute to a scale-dependent interaction with its hyperparameters and examine in Appendix H. With appropriately chosen hyperparameters, we believe that GRAM would yield a Retain (Virology) Compute Ratio similar to filtering and FT-LoRA at 5B, as well as a very slightly higher Compute Ratio on Core data.

## 7. Related Work

**Data filtering for robust capability removal.** O'Brien et al. (2025) evaluates data filtering for robust capability removal in open-weight models. Their approach excludes targeted data during pre-training, achieving capability removal that resists adversarial elicitation. Chen et al. (2025b) shows that data filtering is highly effective in removing harmful information about CBRN weapons from the model. Recent work showed that data filtering is effective at removing undesired characteristics such as toxicity from the model (Birhane et al., 2023; Longpre et al., 2024; Li et al., 2025; Stranisci & Hardmeier, 2025). However, data filtering requires pre-training compute costs that scale linearly with the number of supported capability profiles.

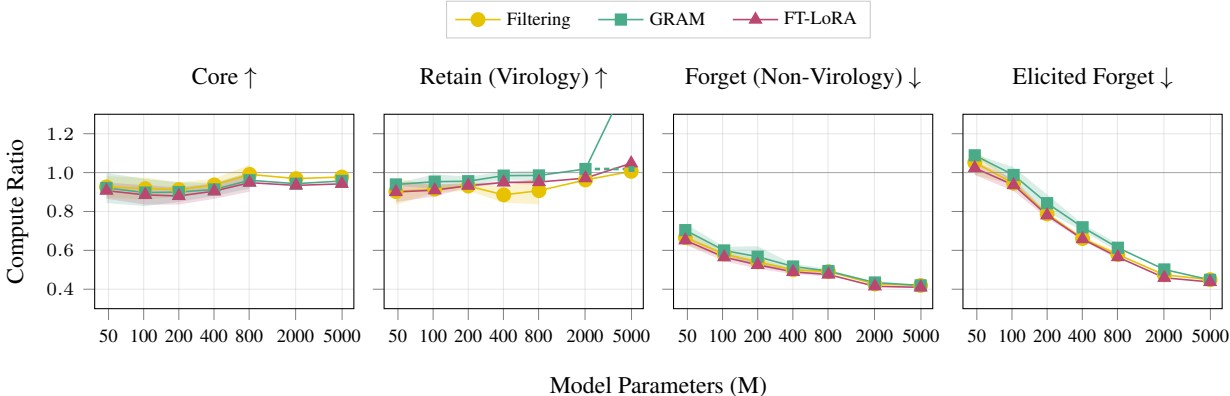

*Figure 7.* **GRAM and FT-LoRA approximate data filtering performance across scales.** Compute ratio versus model size for GRAM, FT-LoRA, and filtering, for the configuration that retains only core and virology. Larger models are trained on larger datasets according to Chinchilla-optimal scaling. Both GRAM and FT-LoRA closely track the capability profile of the data filtered model, despite only requiring a single training run. For all methods, the relative degree of forgetting and the robustness to adversarial finetuning grows as a function of scale. For models below 2B parameters, the shaded region indicates 90% CIs for the mean over $N = 3$ independent training runs. Only one run was done for sizes 2B and above. *Note: the Retain performance of GRAM at 5B params is anomalous; we discuss in the text and Appendix H, and indicate our guess at the correct value with a dotted line.*

**Training for modularity.** There are a variety of methods to induce modularity in neural networks. Modular deep learning imposes architectural structure to decompose learning into task-specific modules (Pfeiffer et al., 2023), constraining how models organize computation. In contrast, GRAM only controls which parameters receive gradient updates from which data, placing no constraints on the representations learned within each partition. In mixture-of-experts architectures (Jacobs et al., 1991; Shazeer et al., 2017), routing functions can be supervised using metadata such as language (Pfeiffer et al., 2022) or domain (Gururangan et al., 2022). DEMix (Gururangan et al., 2022) assigns each expert to a domain and activates only the corresponding expert for domain-specific data, but its symmetric expert sizing and lack of a shared core limit performance under class imbalance.

In concurrent work, Ghosal et al. (2026) proposes a method that localizes knowledge into ablatable parameters, similar to GRAM. This method uses sparse masks to target specific facts for later removal. A simple but potentially important methodological difference is that GRAM restricts gradients from forget data from reaching shared parameters (via $p_{as}$) with the aim of achieving better isolation of forget capabilities. Experimentally, we use compute-optimal scaling to evaluate isolation of a small number of broad capabilities, whereas Ghosal et al. (2026) isolates a large number of individual facts at a single model size.

Modularity can also be induced without supervision. Wang et al. (2024) observe that routing distributions in standard MoE models are highly concentrated, enabling selective finetuning of task-relevant experts; however, this relies on emergent specialization in learned routers, which does not produce reliable domain-level separation (Xue et al., 2024). Golechha et al. (2025) train with a clusterability loss that encourages non-interacting clusters, yielding smaller circuits, though they report that this does not produce increased task specialization. In contrast to these unsupervised approaches, GRAM extends supervised routing by combining it with asymmetric module sizing and an always-active core, enabling robust capability removal via module ablation.

**Continual learning.** Continual learning methods learn from data that arrives sequentially. Such methods often seek to limit interference between tasks by reserving a subset of weights for each task, growing a neural network as new tasks arrive, or conditioning a shared network on a small per-task input such as a learned prompt or task embedding (Mallya & Lazebnik, 2018; Mallya et al., 2018; Serra et al., 2018; Rusu et al., 2016; Yoon et al., 2018; Aljundi et al., 2017; von Oswald et al., 2020; Razdaibiedina et al., 2023; Bohao et al., 2024). Our setting differs in that all data is available during training; the goal is to produce multiple capability profiles from a single pre-trained model rather than to acquire new tasks sequentially without forgetting old ones. The most relevant methods from this literature are therefore continued pre-training (Xie et al., 2024; Guo et al., 2025) and parameter-efficient adaptation via LoRA (Hu et al., 2022; Lialin et al., 2023; Han et al., 2024), both of which we evaluate as baselines.

## 8. Discussion

**Capability control involves competing objectives; no method dominates.** Methods for capability control must trade off core performance, retain performance, forget performance, robustness to forget elicitation, training cost, in-

ference cost, memory requirements, and implementation complexity. These objectives cannot be reduced to a single metric, and which to prioritize depends on the use case. Our experiments aim to evaluate methods as fairly as possible, holding compute cost fixed and accounting for total parameters. We discuss compute-equivalent comparisons in Appendix I.

**Why do single-run methods match data filtering?** It is not obvious that GRAM, FT-LoRA, and FT-Full should approximate data filtering as closely as they do (Figure 4), since the four methods utilize the auxiliary data very differently. Data filtering trains each model on only one auxiliary domain, whereas GRAM is trained over all domains; this appears to improve GRAM's performance on retained auxiliary capabilities through cross-domain transfer. The branched methods (FT-LoRA, FT-Full) differ by introducing auxiliary domain data in a separate finetuning phase rather than mixing the data throughout pre-training.

Prior work finds that introducing a domain through later, sequential training deteriorates capabilities from pre-training, but mixing it in from the start avoids this (Guo et al., 2025). This suggests FT-LoRA should have lower core performance than data filtering, yet it remains competitive with scale: in our setting, the differences between separating, ordering, and mixing data are small. With a larger proportion of auxiliary data, multi-epoching, or different capability profiles, these methods may diverge from data filtering and from one another.

**Architectural separation enables robust removal.** Prior work shows it is difficult to remove a capability once a model has learned it (Lee et al., 2025). Data filtering and model branching (e.g. FT-LoRA) avoid this by never training the model to have the capability in the first place. Post-hoc unlearning fails to overcome it, as shown in the literature cited earlier and in our 800M experiments, where MaxEnt's unlearning is mostly reversed by finetuning. In GRAM and FT-LoRA the capability is instead localized to a dedicated module whose contribution can be continuously scaled at inference: scaling a single module's forward weight from zero to one restores its capability up to baseline level while leaving core performance unchanged (Appendix N). A per-token analysis (Appendix L) illustrates this localization by demonstrating how the increased loss induced by ablating the module associated with a capability concentrates on a sparse set of domain-specific tokens.

**Partial labeling reverses the usual ordering.** With fully labeled data, filtering is the gold standard for removal; with half the labels missing, GRAM removes capabilities more effectively (Section 5). Filtering and FT-LoRA must treat unlabeled data as core, so unlabeled auxiliary examples update the shared parameters and the capability is learned anyway. GRAM trains on the same unlabeled data with all modules active, yet its forget performance degrades far less. We attribute this to the absorption effect observed in prior gradient routing work (Cloud et al., 2024; Shilov et al., 2025): once labeled examples cause a module to specialize on a domain, updates from similar unlabeled examples concentrate in that module, so capability learned from unlabeled data accumulates disproportionately in ablatable parameters rather than in the core.

**Modular capability restriction could enable safe, flexible deployment of highly capable AI systems.** Grounding access control in capability restriction rather than behavioral steering or classification grants stronger assurances against jailbreaks and misuse. GRAM or FT-LoRA could enable model providers to serve dual-use capabilities only in authorized deployments, all from a single model, by controlling which modules are active at inference time. GRAM or FT-LoRA may also be relevant for open-weight releases, where ablating dual-use modules before distribution would allow for model releases with purely beneficial capabilities.

**Limitations.** (i) Any method that achieves access control via capability shaping faces the challenge of entangled capabilities. When target capabilities overlap substantially with desired ones, clean separation may be impossible regardless of method. (ii) Our scaling experiments show favorable trends up to 5B parameters, but we have not verified whether these trends continue at frontier scales or hold in a production setting. (iii) Implementing GRAM in production would require significant effort and introduce complexity that frontier AI developers may prefer to avoid. (iv) We do not yet have a clear account of why GRAM composes capabilities more reliably than FT-LoRA, whether the absorption account of GRAM's partial-labeling advantage holds up under direct investigation, or whether either advantage persists beyond the regimes we study. (v) Nor do we know how either method interacts with post-training such as instruction tuning or RLHF, which could either reinforce or erode the capability separation established during pre-training. (vi) Finally, we evaluate performance using cross-entropy loss, which is a strong predictor of downstream performance (Hoffmann et al., 2022; Gadre et al., 2025; Chen et al., 2025a), but may not give a complete picture.

## 9. Conclusion

As AI capabilities grow, so does the need for robust access control. Gradient-routed auxiliary modules (GRAM) may be a step toward a solution, enabling the equivalent of many data-filtered models to be trained for the cost of a single run. Importantly, our results establish that data filtering itself scales favorably, and that both GRAM and FT-LoRA track it closely across scales. Of the two, GRAM additionally supports composable capabilities and performs better under partial labeling, though these benefits need further study.

## Acknowledgements

We gratefully acknowledge Igor Shilov, Jacob Goldman-Wetzler, Neil Rathi, Nina Panickssery, Jake Ward, Boyd Kane, Kyle O'Brien, Cam Tice, Puria Radmard, Edward Young, Nathalie Kirch, Alex Infanger, Mikita Balesni, David Africa, Alex McKenzie, the anonymous reviewers, and unnamed others for helpful feedback on earlier drafts. We thank Ryan Greenblatt and Krishna Patel for inspiring conversation.

We thank Mojmir Stehlik, John Hughes, Ethan Perez, Ryan McConnell, Kai Huang, Jared Brown, and Anthropic for providing compute support for the scaling experiments.

## Impact Statement

We study methods for controlling access to model capabilities in large language models, motivated by risks posed by dual-use capabilities. This work could help developers reduce misuse while enabling them to deploy capable models more broadly, including to beneficial applications.

We hope that this work is a step toward safer deployment of increasingly capable models, and we encourage future work to study the limitations of such approaches, their interaction with adversarial users, and their integration with organizational and policy-level safeguards.

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

# APPENDIX TO MODULAR PRETRAINING ENABLES ACCESS CONTROL

## A. Details for the Simple Stories Results

**Dataset and auxiliary categories.** Simple Stories (Finke et al., 2025) is an LLM-generated dataset of approximately 2 million short children's stories, each annotated with one of 48 high-level topic categories. These labels enable a clean partition of the training distribution into capability-aligned subsets. We designate four auxiliary capabilities corresponding to the first four categories in alphabetical order: *A Deadline or Time Limit*, *Alien Encounters*, *Bygone Eras*, and *Cultural Traditions*. All remaining 44 categories are pooled to form the core dataset $\mathcal{D}_1$.

**Model architecture.** All models use a decoder-only Transformer architecture with 8 layers, 8 attention heads per layer, embedding dimension 512, and sequence length 256. We use the `SimpleStories/SimpleStories-1.25M` tokenizer with a vocabulary size of 4096 tokens. Unfiltered baseline and comparison models utilize a standard feedforward dimension of 2048, yielding approximately 26M trainable parameters.

For GRAM, each Transformer MLP block is modified via the addition of four auxiliary modules. The core MLP uses a hidden dimension of $d_{\text{core}} = 1856$, while each auxiliary module is itself an MLP with a reduced dimension of $d_{\text{aux}} = 192$. Each auxiliary module contains approximately 1.6M parameters, corresponding to roughly 6% of the total model capacity. We train both GRAM and baseline over all available data. We ensure approximate training compute equivalence of GRAM with an unfiltered baseline model by ensuring GRAM's maximum active parameter count never exceeds the total parameter count of the baseline model.

**Training procedure and optimization.** All models are trained for a single epoch with batch size 128 and a fixed context length of 256 tokens. We use the AdamW optimizer with $\beta_1 = 0.9$, $\beta_2 = 0.95$, and weight decay 0.1. The learning rate is set to $5 \times 10^{-3}$ with a warmup-stable-decay (WSD) schedule: linear warmup for 10% of training steps, constant learning rate for 80%, and linear decay for the final 10%. Gradients are clipped to a maximum $\ell_2$ norm of 1.0.

Training is performed in bfloat16 precision, and all runs use fixed random seeds for reproducibility. The same optimizer, learning rate schedule, and number of training steps are used across all baselines and GRAM variants to ensure compute parity.

**Description of compared methods.**

- **GRAM**: Gradient-routed auxiliary modules, where data-conditioned gradient updates localize capabilities into distinct modules during pre-training. We set $p_{\text{as}} = 0.3$ and $p_{\text{cr}} = 0.5$, and train with heterogeneous accumulation (Appendix H).

- **Baseline**: A standard Transformer trained on all data $\bigcup_{i=1}^{N} \mathcal{D}_i$.

- **Filtering**: For each auxiliary capability $i$, we train a separate model on $\mathcal{D}_1 \cup \mathcal{D}_i$, as well as an additional model trained only on $\mathcal{D}_1$. This approach provides a gold standard for robust capability removal but requires $N$ independent training runs.

- **MaxEnt** (Yuan et al., 2025): MaxEnt is a post-hoc machine unlearning method that modifies a trained language model to suppress unwanted capabilities without retraining from scratch. The method operates by explicitly maximizing the model's output entropy on the forget set while preserving performance on the retain set.

  For each capability profile, we define $\mathcal{D}_{\text{retain}}$ as the core plus retained auxiliary datasets and $\mathcal{D}_{\text{forget}}$ as the remaining auxiliary datasets, initialize from the same pre-trained baseline, and run MaxEnt for 2000 steps. The main parameter of MaxEnt is the retain-loss weight, which scales the retain-set cross-entropy loss relative to the entropy-maximization loss on the forget set. We tune this weight and the learning rate per profile via Bayesian optimization.

- **DEMix** (Gururangan et al., 2022): Modular architecture with domain-specific MLP experts of equal size, designed to encourage domain specialization. Unlike GRAM, DEMix lacks a larger always-active core MLP, has shared parameters updated by all data categories, does not support multiple active experts, and does not allow for stochastic updating of experts on categories unrelated to their specialization.

- **FT-Full**: We first train a base model on a fraction of the core dataset $\mathcal{D}_1$ and branch the model into N versions, each of which is then subsequently trained on a mixture of data for core and a single auxiliary class. The number of training steps used for any given model is controlled such that the sum of all training steps across all models is equal to the

training steps used for the baseline model. The core–auxiliary ratio $p_{ca}$ sets the mix of core and auxiliary data during finetuning; all experiments in this section use $p_{ca} = 2.0$.

- **FT-LoRA** (Hu et al., 2022): This method is analogous to FT-Full, but where the finetuning is performed via a low-rank adapter instead of full-rank finetuning. All experiments in this section use $p_{ca} = 2.0$.

All models are trained with three random seeds, and we report mean performance with confidence intervals across runs.

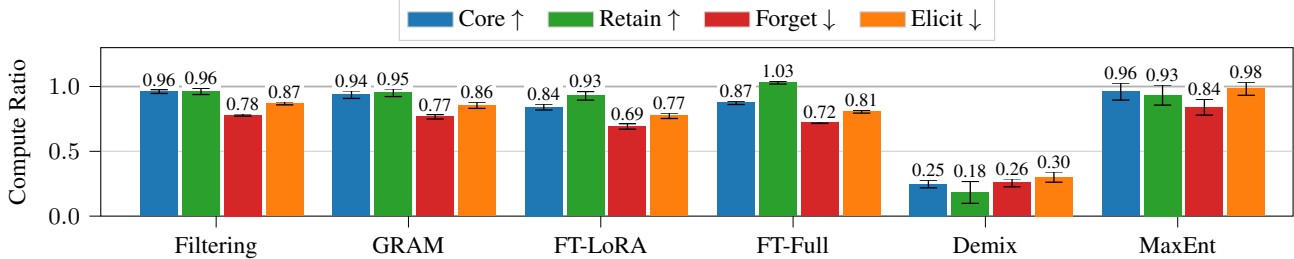

*Figure 8.* **Aggregate compute ratios on Simple Stories.** Compute ratio for each method, averaged over capabilities within each metric class (core, retain, forget, and elicited forget); the visual companion to Table 1. Higher is better for Core and Retain; lower is better for Forget and Elicited Forget. Error bars show 90% CIs for the mean over $N = 3$ independent training runs.

*Table 1.* Compute ratio results on Simple Stories (mean (error)).

| Method | Core | Retain | Forget | Elicited Forget |
|---|---|---|---|---|
| GRAM | 0.938 (0.029) | 0.952 (0.030) | 0.766 (0.017) | 0.855 (0.024) |
| Filtering | 0.961 (0.015) | 0.962 (0.024) | 0.780 (0.009) | 0.870 (0.012) |
| FT-LoRA | 0.841 (0.023) | 0.928 (0.033) | 0.692 (0.021) | 0.774 (0.021) |
| FT-Full | 0.874 (0.013) | 1.029 (0.011) | 0.722 (0.007) | 0.806 (0.011) |
| MaxEnt | 0.961 (0.066) | 0.934 (0.078) | 0.840 (0.061) | 0.983 (0.051) |
| DEMix | 0.248 (0.030) | 0.183 (0.085) | 0.257 (0.031) | 0.302 (0.040) |

### A.1. Extended Simple Stories Results

**Evaluation metrics.** We evaluate models using retain, core, and forget performance, all reported via the compute ratio metric defined in Section 2. *Retain and core performance* measure how well a model preserves desired capabilities. We compute $\ell_R = \frac{1}{|\mathcal{R}|} \sum_{i \in \mathcal{R}} \ell(\mathcal{M}, \mathcal{D}_i)$ and $\ell_C = \ell(\mathcal{M}, \mathcal{D}_1)$. Lower loss (higher compute ratio) values show better performance. *Forget performance* measures the extent to which excluded capabilities are removed. $\ell_F = \frac{1}{|\mathcal{F}|} \sum_{i \in \mathcal{F}} \ell(\mathcal{M}, \mathcal{D}_i)$. Lower compute ratio values indicate better capability removal. We first compute the compute ratio for each capability, then average the compute ratio values while reporting the results.

**Results: Capability retention and removal.** Table 1 summarizes the performance tradeoffs across methods. GRAM has a retain compute ratio of **0.95**, closely matching data filtering (**0.96**), while maintaining comparable core performance (**0.94** vs **0.96**). It achieves this with only a single training run and a single deployed model, whereas data filtering requires training and deploying $N$ separate models.

Among single-run approaches, GRAM provides the strongest overall tradeoff. DEMix performs poorly on both retain and core datasets (**0.18**, **0.25**), reflecting the absence of shared parameters. FT-LoRA achieves strong retain performance (**0.93**) but lower core performance (**0.84**), while FT-Full attains strong retain (**1.03**) and core (**0.87**). MaxEnt preserves core and retain (**0.96**, **0.93**) but does not robustly remove the forget capability (**0.84**), which recovers almost completely under elicitation (**0.98**).

**Compute scaling advantage.** The compute advantage of GRAM over data filtering scales linearly with the number of auxiliary capabilities. Data filtering requires training and deploying $N$ separate models at a total cost of $N \times C_{base}$, whereas GRAM trains a single model that supports all configurations by module ablation. This single-model deployment avoids the

training and inference overhead associated with multiple filtered models. FT-Full similarly requires deploying $N$ separate checkpoints despite a comparable training cost to the baseline.

**Results: Robustness to adversarial elicitation.** Post-hoc unlearning methods are not robust to adversarial finetuning. We evaluate whether GRAM resists such elicitation methods. *Elicited forget performance* evaluates whether excluded capabilities recover after finetuning. After computing forget performance, we finetune each model for 75 steps using 128 sequences from each forget category, then recompute loss on forget datasets. Lower compute ratio values indicate more robust removal.

**Results: DEMix falters in class imbalanced settings.** The performance of DEMix significantly lags behind other methods in both core performance (**0.25**) and retain performance (**0.18**). We attribute this behavior to the lack of a commonly active core expert in a class imbalanced setting. Without the ability for auxiliary modules to augment the capabilities of a common core, DEMix experts instead must learn basic understanding of the dataset redundantly in each expert. When an expert is trained on a small fraction of the overall dataset volume, this leads to poor downstream performance.

Table 1 (right two columns) reports the results on forget capabilities after adversarial finetuning. GRAM shows strong robustness, with an elicited forget compute ratio of **0.86**, only modestly above its pre-elicitation forget ratio of **0.77**. Data filtering reaches a similar elicited value (**0.87**), suggesting that GRAM approaches the robustness of not training on the excluded data. In contrast, post-hoc unlearning is substantially less robust: MaxEnt almost fully recovers under adversarial finetuning (**0.98**), indicating that forget capabilities remain encoded in the parameters. FT-Full recovers to **0.81**, FT-LoRA to **0.77**, while DEMix shows minimal recovery (**0.30**) due to poor overall learning.

**Discussion.** These results show that GRAM enables robust capability removal via architectural separation during training. Isolated capabilities resist recovery under adversarial finetuning, while retain capabilities remain largely unaffected. In contrast, post-hoc unlearning methods that modify pre-trained weights do not prevent recovery. GRAM achieves this robustness while supporting multiple capability configurations in a single model, yielding substantial training and deployment efficiency gains.

# B. Details for the Realistic Dual-Use Baseline

**Datasets and capability partitioning.** We evaluate selective capability control in a realistic dual-use setting using a mixture of general-purpose text and four specialized, safety-relevant domains. The *core dataset* $\mathcal{D}_1$ combines FineWeb-Edu (educational web text), general-purpose arXiv papers, and non-Lisp code from The Stack (BigCode, 2023).

The inclusion of web text and code in the core reflects standard pre-training data mixtures used by major language model developers (Touvron et al., 2023; Llama Team, AI @ Meta, 2024). We additionally include a uniform sample of general-purpose arXiv papers, processed through the same pipeline as the auxiliary papers (PDF to markdown conversion, tokenization). This design choice ensures that any measured differences in compute ratio between core and auxiliary domains reflect differences in domain-specific content rather than formatting artifacts. Without this control, lower loss on auxiliary papers could partially reflect the model learning arXiv-specific formatting conventions (e.g., LaTeX structure, citation patterns, section headings) rather than dual-use knowledge. By exposing the core to identically formatted academic text, we isolate the signal attributable to domain content.

We define four *auxiliary capability datasets*, each corresponding to a dual-use domain:

- **Code**: Lisp code from The Stack (BigCode, 2023), serving as a representative for specialized programming tasks, like those involved when modifying an internal codebase.

- **Virology**: Virology papers from Europe PMC.

- **Nuclear**: Papers from OSTI (Office of Scientific and Technical Information) and arXiv.

- **Cybersecurity**: Papers on operating systems, networking, security, and cryptography from arXiv (`cs.CR`, `cs.NI`, `cs.OS`).

Our training data distribution includes 99% general purpose core data, and 1% auxiliary data. We use 16B core tokens for training (sourced from FineWeb-Edu, non-Lisp code, and general-purpose arXiv papers). The auxiliary data collectively is

*Table 2.* **Training data mixture for the dual-use setting.** Token counts correspond to the 800M-parameter runs. The same proportions are held fixed across the scaling analysis in Figure 7.

| Domain | Source | Type | Tokens | % |
|---|---|---|---|---|
| Web text | FineWeb-Edu | Core | 13.4B | 83.2 |
| Code (non-Lisp) | The Stack | Core | 2.4B | 14.9 |
| General papers | arXiv | Core | 160M | 1.0 |
| Code (Lisp) | The Stack | Aux. | 40M | 0.25 |
| Virology | Europe PMC | Aux. | 40M | 0.25 |
| Nuclear | OSTI, arXiv | Aux. | 40M | 0.25 |
| Cybersecurity | arXiv | Aux. | 40M | 0.25 |

drawn uniformly across the four dual-use categories. Capability labels are assigned at the dataset level and used to control module activation and gradient routing during training. Table 2 summarizes the full mixture.

**Model architecture.** All models use a decoder-only Transformer architecture with 24 layers, 8 attention heads with 2 key-value heads (grouped-query attention), embedding dimension 1600, and a feedforward (MLP) dimension of 6400. We use the tokenizer `EleutherAI/gpt-neo-125M` with a vocabulary size of 50,304 tokens, and the maximum context length is 1024 tokens. This configuration yields a model with approximately 800M parameters for the baseline model architecture.

For gradient-routed auxiliary modules (GRAM) models, each Transformer MLP block is augmented by four *auxiliary modules*, each a smaller per-domain MLP. The core MLP has a feedforward dimension ($d_{\text{core}} = 6400$), while each auxiliary module uses a hidden dimension of ($d_{\text{aux}} = 640$), corresponding to 10% of the baseline MLP width. The number of auxiliary parameters per dataset is 49.2M.

**Training procedure and optimization.** All models are trained for a single epoch over the combined dataset using an effective batch size of 672 sequences (micro-batch size 12 per GPU with 7 gradient accumulation steps) and a context length of 1024 tokens. Training is distributed across 8 GPUs using data parallelism. We use the AdamW optimizer with $\beta_1 = 0.9$, $\beta_2 = 0.95$, and weight decay 0.1. The peak learning rate is set to $5.7 \times 10^{-4}$ and follows a warmup-stable-decay (WSD) schedule with linear warmup for 2% of steps, a constant phase, and linear decay over the final 10% of steps. Gradients are clipped to a maximum $\ell_2$ norm of 1.0.

Training is performed in bfloat16 precision, and all runs use fixed random seeds for reproducibility. The same optimizer and learning rate schedule are used across all compared methods. Branched training variants initialize a new learning rate scheduler for each subsequent finetuning phase. The average number of active parameters for GRAM and FT-LoRA is larger than the dense transformer baseline. To ensure compute equality, we downsample training data to keep total training FLOPs constant. More details are provided in Appendix I.

**Adversarial elicitation.** For the elicited forget metric, we finetune a copy of each evaluated model on a fixed sample of 512 sequences of 1024 tokens (approximately 0.5M tokens) drawn from the training split of each forget category, roughly 1.3% of that domain's 40M training tokens at the 800M scale. We train for up to 200 epochs over this sample at one quarter of the pre-training peak learning rate, with early stopping on validation loss, and report the best validation loss reached.

**Description of compared methods.**

- **GRAM**: Gradient-Routed Auxiliary Modules where data conditioned gradient updates localize capabilities into distinct modules during pre-training. We set $p_{\text{as}} = 0.5$ and $p_{\text{cr}} = 0.2$.

- **Baseline**: An unfiltered, standard dense Transformer trained on the full mixture of core and auxiliary data.

- **Filtering**: A standard Transformer trained exclusively on the core dataset plus 0 or 1 auxiliary categories, with reported results aggregated over all $N$ model variants.

- **MaxEnt**: A post-hoc unlearning method applied after pre-training, maximizing output entropy on excluded auxiliary domains. We run MaxEnt for 2000 steps per capability profile, with the learning rate and retain-loss weight tuned per profile via Bayesian optimization over $[5 \times 10^{-5}, 2 \times 10^{-4}]$ and $[50, 600]$, respectively.

- **FT-Full**: A finetuning-based method that first trains a core model on the core dataset and finetunes separate branched

checkpoints on mixtures of core and one auxiliary dataset. All experiments in this section use $p_{\text{ca}} = 4.0$, $p_{\text{af}}(i) = 1.0$, and $p_{\text{lr}} = 0.2$.

- **FT-LoRA**: A similar approach to FT-Full, but where the finetuning is achieved via low-rank adapters applied to a frozen core-only baseline model, with one set of adapters trained for each auxiliary capability. All experiments in this section use $p_{\text{ca}} = 1.0$, with auxiliary factor $p_{\text{af}} = 2.0$ for Code and $1.0$ for the other domains, and $p_{\text{lr}} = 1.0$.

All comparison methods are evaluated using identical validation splits and metrics, enabling direct comparison of retention, forgetting, and robustness to adversarial finetuning.

## C. Scaling Experiment Details

Table 3 gives per-model training details for the experiments in Section 6.

| Model Size | Aux Module Size | # Layers | Core MLP Dim | Aux MLP Dim | Batch Size | Core Data Size | Peak LR |
|---|---|---|---|---|---|---|---|
| 50M | 0.6M | 6 | 1536 | 128 | 72 | 0.96B | $1.3 \times 10^{-3}$ |
| 100M | 3.0M | 9 | 2560 | 256 | 128 | 2.1B | $1.1 \times 10^{-3}$ |
| 200M | 9.0M | 13 | 3584 | 384 | 216 | 4.0B | $8.7 \times 10^{-4}$ |
| 400M | 22.4M | 18 | 4864 | 512 | 368 | 8.0B | $7.0 \times 10^{-4}$ |
| 800M | 49.2M | 24 | 6400 | 640 | 672 | 16.1B | $5.7 \times 10^{-4}$ |
| 2B | 132.2M | 32 | 9216 | 896 | 1329 | 40.3B | $4.3 \times 10^{-4}$ |
| 5B | 351.3M | 47 | 12288 | 1216 | 2752 | 99.4B | $3.2 \times 10^{-4}$ |

*Table 3.* GRAM model and training configurations used in scaling experiments. All models use four auxiliary modules corresponding to Virology, Cybersecurity, Nuclear Physics, and Code. Auxiliary data is 1% of core data. The embedding dimensions are the MLP dimensions divided by four. All models are trained for one epoch using AdamW with a warmup-stable-decay (WSD) learning rate schedule. Peak learning rates are determined by a fitted power law ($0.311 \cdot N^{-0.308}$, $R^2 = 0.99$).

## D. GRAM Scaling with Increasing Numbers of Auxiliary Capabilities

This section examines how GRAM performance scales as the number of auxiliary capabilities increases. The goal is to test whether introducing many auxiliary modules leads to degradation of core performance or collapse of capability separation.

**Experimental Setup.** We conduct a sweep over the number of auxiliary capabilities on the Simple Stories dataset, which contains 48 distinct topic categories. We use Simple Stories rather than the realistic setting here because the latter has only four auxiliary domains, too few to study how performance scales with the number of capabilities. For each configuration, we select $N \in \{4, 8, 12, 16, 20\}$ categories to serve as auxiliary capabilities.

To isolate the effect of increasing $N$, we hold the total training budget and the core/auxiliary token composition fixed across all experiments. For every $N$, the $N$ auxiliary categories together receive 20% of the training tokens while the remaining $48 - N$ categories (the core dataset $\mathcal{D}_1$) receive the other 80%, matching the full-dataset token budget; we up- or down-sample each side as needed to hit this 80%/20% split regardless of how many categories fall on each side. This ensures that changes in performance reflect the number of auxiliary categories rather than variation in the core/auxiliary data balance or total training compute.

For each value of $N$, the auxiliary categories are the first $N$ topics under a fixed alphabetical ordering of the 48 categories, so the auxiliary sets are nested as $N$ grows; the same categories are used for every seed, with seeds varying only training stochasticity (weight initialization and data ordering). Results are averaged across these seeds. All other training details, including model architecture, optimization hyperparameters, and training duration, follow the Simple Stories experiments in the main text, including auxiliary spread $p_{\text{as}} = 0.3$ and core robustness $p_{\text{cr}} = 0.5$.

Importantly, the number of *active parameters per forward pass* is held constant across all configurations. Adding auxiliary capabilities increases the number of dormant module parameter sets, but does not increase per-step compute or effective model capacity during training or inference.

**Results.** Figure 3 shows compute ratio performance as the number of auxiliary categories increases. GRAM maintains near-baseline core performance across all settings, with compute ratios staying roughly constant at around 0.88 from 4 to

20 categories. Retain performance similarly remains stable (approximately 0.89–0.91), indicating that auxiliary modules represent their designated capabilities, even with an increasing number of categories.

In contrast, forget performance remains substantially lower, with compute ratios of approximately 0.65–0.74 across all configurations. This separation persists after adversarial finetuning, where elicited-forget performance (approximately 0.71–0.84) remains below retain performance. We note that forget and elicited-forget performance rise modestly as the number of categories increases—expected, since the fixed 20% auxiliary budget is divided among more topics, leaving less data to isolate each—while core and retain performance stay roughly flat.

Importantly, increasing the number of auxiliary categories does not degrade core performance, nor does it collapse the separation between retain and forget capabilities. These results demonstrate that GRAM scales gracefully with the number of auxiliary modules, enabling robust capability localization within a single training run.

These experiments show that GRAM can support an increasing number of auxiliary capabilities without sacrificing core performance or robustness. Under a fixed compute budget, a single GRAM model approximates the behavior of many separately trained, data-filtered models, while enabling selective and structurally robust capability removal via ablation.

## E. Architecture Ablation: MLP vs. LoRA

Our experiments with real world data contrast two competitive methods: GRAM and FT-LoRA. These methods differ both in training methodology and model architecture: GRAM isolates capabilities into small auxiliary MLPs during pre-training using a specific update rule, LoRA isolates capabilities in auxiliary LoRA adapters during standard finetuning. In this section, we isolate the effect of auxiliary module architecture. We find that performance characteristics depend only on training methodology, not on architecture.

We evaluate the following variants:

- **GRAM (MLP)**: Gradient-Routed Auxiliary Modules implemented via small auxiliary MLPs. This is the variant of GRAM utilized in all other experiments. Training uses $p_{as} = 0.5$, $p_{af}(i) = 1.0$, and $p_{cr} = 0.2$.

- **GRAM (LoRA)**: Identical to GRAM (MLP) but implemented with LoRA adapters instead of additional MLP neurons. Training uses $p_{as} = 0.5$, $p_{af}(i) = 1.0$, and $p_{cr} = 0.2$.

- **FT-LoRA**: Branched training method with auxiliary capabilities are finetuned into a core-only filtered model via low rank adapters. Training uses $p_{ca} = 1.0$, $p_{af}(i) = 1.0$, and $p_{lr} = 1.0$.

- **FT-MLP**: Identical to FT-LoRA but where the LoRA adapters are replaced with small auxiliary MLPs. Training uses $p_{ca} = 1.0$, $p_{af}(i) = 1.0$, and $p_{lr} = 1.0$.

All variants are trained with equal compute and the number of parameters allocated to a given auxiliary capability is fixed to 10% of the core MLP module parameters. All methods use the heterogeneous accumulation methodology from Appendix H and train models with 100M parameters.

**Results.** Figure 9 provides a comparison between the four variants. Within a given training methodology, there is little variation in capability profiles between architectural variants. Between training methodologies, the capability profiles vary more significantly.

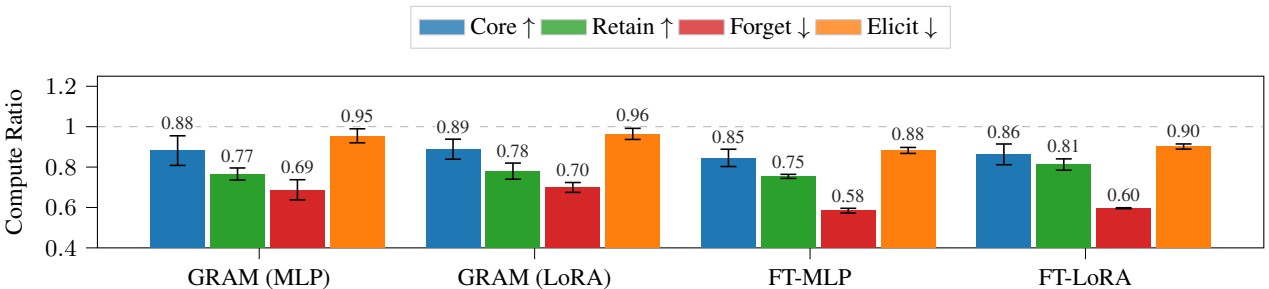

*Figure 9.* Comparison of GRAM via MLP, GRAM via LoRA, branched training via LoRA finetuning, and branched training via auxiliary MLP finetuning. Black bars show 90% CIs for the mean over $N = 3$ independent training runs. Within a given training method, models achieve similar capability profiles regardless of auxiliary module architecture.

## F. Hyperparameter Appendix

GRAM exposes two routing hyperparameters — *auxiliary spread* $p_{as}$ and *core robustness* $p_{cr}$ (Section 3) — which govern the tradeoff between retaining desired capabilities, suppressing excluded capabilities, and keeping core performance stable across the configurations of active modules. FT-LoRA exposes one comparable knob, the *core:auxiliary ratio* $p_{ca}$, which sets the proportion of core and auxiliary data during finetuning. We sweep each knob on Simple Stories, and report compute ratios aggregated over each capability profile (mean within seed, then mean over $N = 3$ seeds; shaded regions are 90% CIs).

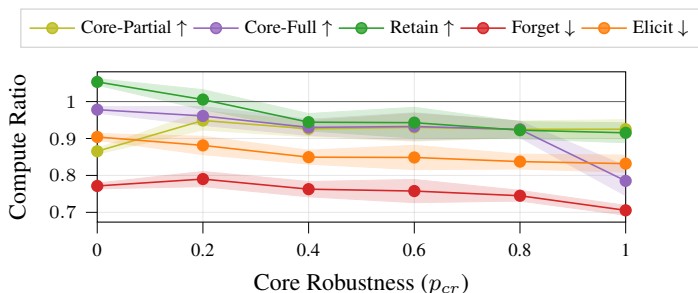

*Figure 10.* **GRAM core robustness sweep** ($p_{cr}$, with $p_{as} = 0.3$). We split the aggregate Core compute ratio (the mean over all five retain configurations) into its two components: *Core-Partial* (yellow) is the mean over the four configurations that retain core plus a single auxiliary module, and *Core-Full* (purple) is the core-only configuration with all auxiliary modules ablated.

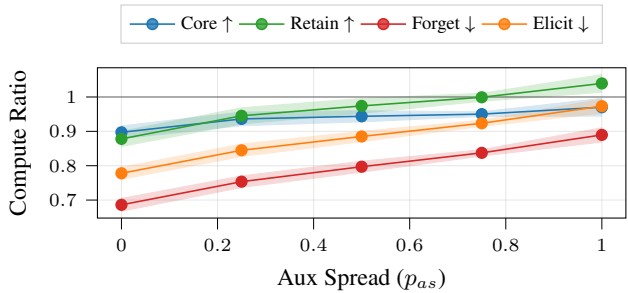

*Figure 11.* **GRAM auxiliary spread sweep** ($p_{as}$, with $p_{cr} = 0.5$). Core is the mean core compute ratio over all five retain configurations.

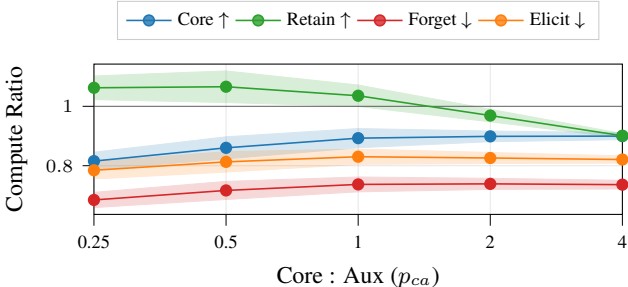

*Figure 12.* **FT-LoRA core:auxiliary ratio sweep** ($p_{\mathbf{ca}}$). Increasing the ratio shifts parameter budget from the auxiliary adapters to the frozen core-only model. Core is the mean core compute ratio over all five retain configurations.

**Auxiliary spread** ($p_{\mathbf{as}}$). For an auxiliary batch from $\mathcal{D}_{i>1}$, the forward pass activates the core MLP and auxiliary module $E_i$; in the backward pass $E_i$ is always updated, while core parameters receive gradients with probability $p_{as}$ (smaller $p_{as}$ yields stronger isolation). With $p_{cr} = 0.5$ fixed, increasing $p_{as}$ raises every metric together (Figure 11): core and retain improve (from 0.90 and 0.88 at $p_{as} = 0$ to 0.97 and 1.04 at $p_{as} = 1$) as auxiliary gradients enrich the shared representation, but forget and elicited-forget rise in step (from 0.69 and 0.78 to 0.89 and 0.97), indicating progressive re-entanglement of the excluded capabilities. Auxiliary spread thus trades isolation for shared-representation quality, with retain remaining above forget across the full range.

**Core robustness** ($p_{\mathbf{cr}}$). For a core batch from $\mathcal{D}_1$, the core MLP is always active; with probability $p_{cr}$ a random auxiliary module is additionally activated, exposing core training to auxiliary perturbations. We report two inference configurations on the core (Figure 10): *Core-Partial*, averaged over the configurations that retain core plus a single auxiliary module, and *Core-Full*, the core-only configuration. At $p_{cr} = 0$ the two diverge (0.98 ablated vs. 0.87 with an auxiliary module active): auxiliary modules never see core data, so the core performs best when they are removed. Increasing $p_{cr}$ exposes auxiliary modules to core batches and the configurations converge through the mid-range ($\approx 0.93$ each for $p_{cr} \in [0.4, 0.8]$), stabilizing core behavior across module subsets. Retain declines modestly over the sweep ($1.05 \rightarrow 0.92$) as auxiliary specialization weakens, while forget and elicited-forget improve slightly ($0.77 \rightarrow 0.71$ and $0.90 \rightarrow 0.83$), confirming that core robustness stabilizes configurations without re-entangling excluded capabilities. At the extreme $p_{cr} = 1$, however, the core comes to depend on auxiliary activations and the ablated configuration degrades sharply (0.79); we therefore prefer moderate values in practice.

**FT-LoRA core:auxiliary ratio** ($p_{\mathbf{ca}}$). $p_{ca}$ reallocates a fixed parameter budget between the core-only model and each auxiliary adapter. Increasing it favors the core: core performance rises modestly ($0.82 \rightarrow 0.90$) while retain falls ($1.06 \rightarrow 0.90$) as the adapters lose capacity (Figure 12). Forget and elicited-forget remain roughly flat ($\approx 0.69$–$0.74$ and $\approx 0.79$–$0.83$). Unlike GRAM's $p_{as}$, $p_{ca}$ provides no mechanism to trade off forget against retain; it only shifts capacity between core and retained auxiliary performance.

**Practical settings.** Moderate GRAM settings (e.g. $p_{as} \in [0.3, 0.5]$ and $p_{cr} \in [0.2, 0.5]$) provide a favorable balance: near-baseline core and retain performance while keeping forget and elicited-forget compute ratios low. Our main Simple Stories comparison uses $p_{as} = 0.3$ and $p_{cr} = 0.5$, while the realistic experiments use $p_{as} = 0.5$ and $p_{cr} = 0.2$.

## G. Partial Labeling Details

Practical pre-training corpora are rarely labeled exhaustively by capability domain. We test whether GRAM's modularization advantage survives when only half of training tokens carry their category label.

**Description of compared methods.**

- **Core-only filtering**: A single filtered model where labeled training data come only from the core dataset $\mathcal{D}_1$.

- **GRAM**: Gradient-Routed Auxiliary Modules, $p_{as} = 0.5$, $p_{cr} = 0.2$, and $p_{af} = 4$ for *code-lisp* and $p_{af} = 3$ for all others.

- **FT-LoRA**: Branched training via low-rank adapters per auxiliary capability, $p_{ca} = 1.0$, and $p_{af} = 2$ for *code-lisp* and $p_{af} = 1$ for all others.

**Experimental setup.** The dataset, tokenizer, and base architecture follow the realistic configuration described in Appendix B, scaled to 400M parameters per the corresponding entry in Table 3. The primary deviation is removing 50% of data labels: each training shard is partitioned into a *labeled* and *unlabeled* half. The labeled portion is handled as normal per the method implementation. For filtering and FT-LoRA, unlabeled data is treated as belonging to the core dataset, regardless of the true label. For GRAM, unlabeled data follows a modified methodology: when the label of the current batch is unknown, all parameters (core and every auxiliary module) participate in both the forward and backward passes.

We find it necessary to modify the architecture of GRAM in this setting to achieve good performance. We make the following modifications: (1) The core MLP is sized to 95% of the baseline's MLP (2) Each auxiliary module is a small MLP sized to 1.25% of the baseline's MLP

We train three seeds per method, with identical compute budgets, optimizer settings, and labeled/unlabeled splits across methods. We set all methods in these partial labeling experiments to use the heterogeneous accumulation method described in Appendix H. We only consider model configurations that correspond to exclusively retaining core and ablating all auxiliary capabilities.

**Results.** Figure 6 reports compute ratios averaged across three seeds. GRAM achieves a forget compute ratio of **0.60**, substantially below partially labeled filtering (**0.85**) and FT-LoRA (**0.85**), while maintaining a competitive core performance (**0.96** vs **0.99**). All methods outperform the reference perfectly labeled filtering on core (**0.92**), likely due to transfer learning effects. Figure 13 expands the main-text comparison with this perfectly labeled filtering (100%) reference alongside the three partially labeled methods.

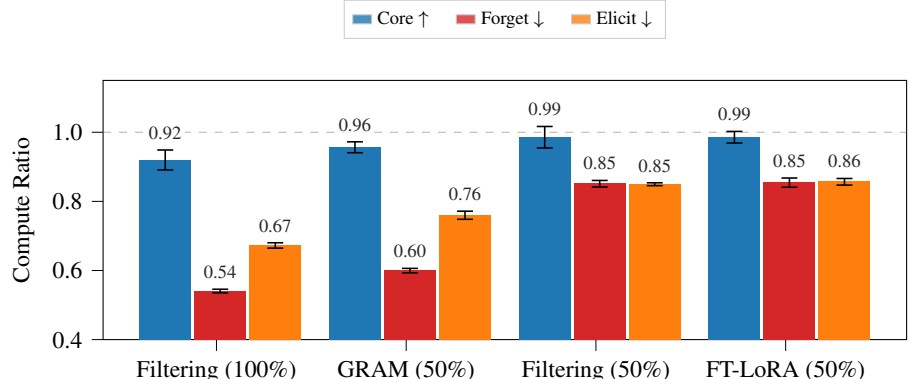

*Figure 13.* **Partial labeling, with the perfectly labeled filtering reference.** The three partially labeled methods (GRAM, filtering, FT-LoRA, all with 50% of tokens labeled) alongside perfectly labeled filtering (100%). Error bars show 90% CIs for the mean over $N = 3$ independent training runs.

**Discussion.** Results indicate the isolation capabilities of GRAM significantly outperform both filtering and FT-LoRA in the partially labeled setting. It is particularly striking that GRAM produces better forgetting than filtering, typically considered the gold standard for unlearning performance. Consistent with Cloud et al. (2024); Shilov et al. (2025) we hypothesize the advantage in GRAM's favor is driven by the *absorption effect*, an empirical observation that the modularity induced by gradient routing persists even when there is no explicit mechanism enforcing gradient isolation for a large portion of training. We leave further investigation of this phenomenon to future work.

# H. Heterogeneous Accumulation

When training with gradient accumulation, each optimizer step aggregates gradients across multiple micro-batches before updating parameters. In our implementation, each micro-batch is associated with a single routing decision: a triple (`label`, `fwd_params`, `bck_params`) specifying which data class is being trained, which parameters participate in the forward pass, and which parameters should receive gradient updates.

Intuitively, gradient accumulation simulates a large batch cheaply: we run several micro-batches, sum their gradients, and take a single optimizer step. The complication for GRAM is that different micro-batches are routed to update different modules, so each module's summed gradient must come only from the micro-batches assigned to it. When all micro-batches in a window route the same way, this is automatic. When they differ, gradients from micro-batches that should not have

touched a module leak into its update and, once summed, cannot be separated out. The rest of this section explains how we ensure each module's accumulated gradient contains only contributions from micro-batches routed to it.

**Uniform accumulation.** When all micro-batches in an accumulation window share the same routing triple, every micro-batch agrees on which modules should be stepped. It is then safe to let `.grad` accumulate indiscriminately across micro-batches and gate the update at window granularity, stepping only the optimizers for modules listed in `bck_params` and zeroing all others.

**Heterogeneous accumulation.** When micro-batches within a window may carry different routing triples, this no longer holds. After shuffling the per-micro-batch routing decisions, a window of $K$ micro-batches might contain a mix of routed-auxiliary batches (updating core and one auxiliary module), unrouted-auxiliary batches (updating only the auxiliary module), and core batches (updating only core parameters). This heterogeneity is necessary to match the target data distribution within each accumulation window rather than only in aggregate across windows.

**Gradient mixing across micro-batches.** The uniform "accumulate then gate" approach breaks once a window mixes routing decisions. Gradients from every micro-batch sum into the same `.grad` buffer, so each module's accumulated gradient picks up contributions from micro-batches that were not meant to update it, and these cannot be separated out at step time. This happens in both directions, but the two are not equally harmful. The core receiving gradients from auxiliary batches is largely benign: the core is large, and auxiliary spread already updates it on a substantial fraction of auxiliary batches by design. The harmful direction is auxiliary modules receiving gradients from core batches. Auxiliary data is only 1% of training, so each module's domain signal is already small, and mixing in core-batch gradients dilutes it further. The only other way to keep gradients separate is to step after each micro-batch, but that is gradient accumulation with a window of one, which defeats its purpose: we use accumulation precisely to train with a large effective batch.

**Freezing modules per micro-batch.** For each micro-batch, only the modules named in its `bck_mask` should receive gradients. Every other module still needs to run in the forward pass, so the output is unchanged, but it must contribute nothing to its own `.grad`. We achieve this by freezing the modules that should not be updated: we run their forward pass on detached copies of their parameters, so no gradient flows back to the real parameters. With the unwanted modules frozen, each module's accumulated gradient at the end of the window comes only from the micro-batches that asked to update it.

We implement freezing with a `freeze()` function that wraps a module's forward pass using `torch.func.functional_call` on detached parameter views:

```
def freeze(module, do_freeze):
  if not do_freeze:
      return module
  params_and_bufs = {
      n: p.detach()
      for n, p in module.named_parameters()
  }
  params_and_bufs.update(
      dict(module.named_buffers())
  )
  def call(*args, **kwargs):
      return torch.func.functional_call(
          module, params_and_bufs, args, kwargs
      )
  return call
```

Detaching shares storage with the original parameter, so the forward output is numerically identical and gradients still flow back through the inputs. But because the detached tensor is not a leaf in the autograd graph, no gradient accumulates on the parameter itself, and the real parameter's `.grad` is never touched.

We apply `freeze()` to each module on every micro-batch according to its `bck_mask`:

```
expert = freeze(
  self.experts[i],
  do_freeze=not bck_mask[i]
)
out = expert(x)
```

At the end of the window, each module's `.grad` holds contributions only from the micro-batches that included it. We then step the optimizers for exactly the modules that received gradient:

```
exp_to_step = [
  label for label in data_labels
  if any(p.grad is not None
         for p in model.get_params(label))
]
```

**Application to scaling experiments.** All 5B configurations reported in the scaling experiments (Section 6) were trained with heterogeneous accumulation using the `freeze()`/`functional_call` mechanism described above. For smaller scales in the scaling experiments, all methods beside the baseline (which is always heterogeneous) train via the uniform accumulation methodology.

Uniform accumulation lowers retain compute ratios relative to heterogeneous accumulation, and the drop is largest for GRAM. We attribute GRAM's larger drop to dilution of its auxiliary updates: the gradient each auxiliary module receives is mixed with contributions from core micro-batches, weakening its per-domain signal. We suspect this is specific to GRAM because of its core-robustness training, though we have not confirmed the mechanism. The gap between uniform and heterogeneous accumulation grows with the number of accumulation steps, and is therefore largest at the biggest model sizes, which use the most steps.

The switch from uniform to heterogeneous accumulation at 5B also explains the off-trend improvement in GRAM's retain performance visible in Figure 7. GRAM runs below 5B use uniform accumulation, while the 5B run uses heterogeneous accumulation; the baseline is heterogeneous at every scale. At the smaller scales, we raised GRAM's auxiliary factor ($p_{\text{af}} = 3.0$ or $4.0$) to offset uniform accumulation's suppression of retain performance and approximate data filtering. At 5B, heterogeneous accumulation removes that suppression, but we did not lower the auxiliary factor to match, leaving retain performance higher than intended. We now believe that with heterogeneous accumulation, GRAM can approximate data filtering at an auxiliary factor of 1.0.

**200M existence proof.** To check that this mechanism matters in practice, we compare uniform and heterogeneous accumulation at 200M for the baseline, GRAM, and FT-LoRA, with every compute ratio measured against the *heterogeneously-trained* baseline of the same size (single seed, auxiliary factor 1.0; Table 4). For GRAM, heterogeneous accumulation is what brings core and retain up to baseline level (core $0.94 \rightarrow 0.99$, retain $0.75 \rightarrow 1.11$) while leaving forget essentially unchanged ($0.70 \rightarrow 0.69$); under uniform accumulation GRAM falls below even the uniformly-trained baseline on retain ($0.75$ vs. $0.76$). FT-LoRA changes little across the two regimes (retain $0.83 \rightarrow 0.90$), indicating the effect is specific to GRAM.

| Method | Accumulation | Core | Retain | Forget |
|--------|--------------|------|--------|--------|
| Baseline | Uniform | 0.95 | 0.76 | — |
| Baseline | Heterogeneous | 1.00 | 1.00 | — |
| GRAM | Uniform | 0.94 | 0.75 | 0.70 |
| GRAM | Heterogeneous | 0.99 | 1.11 | 0.69 |
| FT-LoRA | Uniform | 0.95 | 0.83 | 0.61 |
| FT-LoRA | Heterogeneous | 0.94 | 0.90 | 0.61 |

*Table 4.* **Heterogeneous accumulation at 200M.** Compute ratio for core, retain, and forget evaluations, all measured against the *heterogeneously-trained* baseline of the same size (single seed, auxiliary factor 1.0); the heterogeneous baseline is therefore 1.0 by construction. Heterogeneous accumulation raises GRAM's core and retain to baseline level without changing forget performance, while FT-LoRA changes little across the two regimes, indicating the effect is specific to GRAM.

**Limitations.** Beyond the 5B configurations and the 200M comparison above, we did not run a full heterogeneous-accumulation sweep across model sizes. The 400M and 800M models were trained under uniform accumulation. Because uniform accumulation disproportionately suppresses GRAM's retain performance, these mid-scale GRAM numbers are likely slight underestimates. We expect the effect to be small, since it grows with the number of accumulation steps and these runs use few.

# I. Equal-Compute Training Length Adjustment

In our primary experiments, both GRAM and FT-LoRA have more active parameters than a dense baseline. To ensure compute-matched comparisons, we adjust training lengths so that total training FLOPS equal those of the baseline.

**Two-mode structure.** Both GRAM and FT-LoRA can be thought of as having two modes: a core-only mode (batches drawn from core data with only the core MLP active) and an auxiliary mode (batches drawn from auxiliary data with the core MLP and one auxiliary parameter subset active). For FT-LoRA, these modes occur sequentially; for GRAM they are interleaved throughout training.

**FLOP accounting.** Let $P_{\text{base}}$ denote the number of baseline (dense) parameters, $P_{\text{core}}$ the number of core parameters in the model, and $P_{\text{aux}}$ the average number of parameters in a single auxiliary parameter subset. Let $L_{\text{base}}$, $L_{\text{core}}$, and $L_{\text{aux}}$ denote training lengths (in tokens) for the baseline, core mode, and auxiliary mode respectively. Approximating FLOPS as proportional to tokens $\times$ active parameters:

$$F_{\text{base}} = L_{\text{base}} \cdot P_{\text{base}} \tag{1}$$

$$F_{\text{core}} = L_{\text{core}} \cdot P_{\text{core}} \tag{2}$$

$$F_{\text{aux}} = L_{\text{aux}} \cdot (P_{\text{core}} + P_{\text{aux}}) \tag{3}$$

**Length adjustment.** Total routed FLOPS are $F_{\text{routed}} = F_{\text{core}} + F_{\text{aux}}$. To match the baseline budget, we scale both training lengths by a common factor

$$\lambda = F_{\text{base}} \,/\, F_{\text{routed}},$$

giving adjusted lengths $\lfloor \lambda L_{\text{core}} \rceil$ and $\lfloor \lambda L_{\text{aux}} \rceil$, where $\lfloor \cdot \rceil$ denotes rounding to the nearest integer. This matches total FLOPS exactly up to rounding and preserves the core/auxiliary token mixture: each mode receives additional compute in proportion to its share of $F_{\text{routed}}$, converted to tokens at that mode's active parameter count. When $\lambda > 1$, the routed model trains longer to compensate for having fewer active parameters per step; when $\lambda < 1$, training is truncated. Within the auxiliary mode, the split between auxiliary-data batches and core-data batches (controlled by $p_{\text{ca}}$) is held fixed under rescaling.

**LoRA configuration.** FT-LoRA adapters are matched in parameter count to GRAM auxiliary modules: each per-domain adapter is sized so that its parameter count equals that of one GRAM auxiliary module (`aux_param_prc = 0.1`, i.e. 10% of the baseline MLP width per domain, for $4 \times 10\% = 40\%$ total extra MLP parameters across the four auxiliary domains). Rank is computed dynamically from this parameter budget by equating LoRA parameters to auxiliary GRAM module parameters per block:

$$\text{rank} = \text{round}\left( \frac{2\,d_{\text{emb}}\,d_{\text{aux}} + d_{\text{aux}} + d_{\text{emb}}}{2\,(d_{\text{emb}} + d_{\text{mlp}})} \right),$$

which yields $r \approx 102$ at 400M and $r \approx 128$ at 800M. Adapters are applied to the two MLP linear projections only (the up- and down-projections, `c_fc` and `c_proj` in our code); attention is left unadapted. Adapters are trained in a second, branched phase from a core-only base model; the fraction of compute spent in the core-only phase is controlled by $p_{\text{ca}}$.

# J. Learning Rate and Batch Size Scaling

Across model sizes, we set the peak learning rate (LR) and effective batch size (BS) using power laws fitted from per-size grid searches, rather than re-tuning hyperparameters at every scale. This appendix documents the fitting methodology.

**Grid search.** For each of four model sizes (50M, 100M, 200M, 400M), we train baseline models across a grid of (LR, BS) pairs spaced as powers of two, using three random seeds per configuration. The objective is a weighted cross-entropy loss computed as 50% core CE loss plus 50% mean CE loss across the four auxiliary categories.

**Per-size quadratic fit.** For each model size, we pool results across seeds and fit a quadratic surface in $(\log \text{LR}, \log \text{BS})$ space. The minimum of this surface yields the per-size optimum $(\text{LR}^*, \text{BS}^*)$. Figure 14 displays fit results.

**Cross-size power-law fit.** The per-size optima are fitted against parameter count $N$ in log-log space via least-squares regression:

$$\text{LR}^*(N) = a_{\text{LR}} \cdot N^{\alpha_{\text{LR}}} \tag{4}$$

$$\text{BS}^*(N) = a_{\text{BS}} \cdot N^{\alpha_{\text{BS}}} \tag{5}$$

The fitted coefficients are:

|  | Coefficient $a$ | Exponent $\alpha$ | $R^2$ |
|---|---|---|---|
| Learning Rate | 0.311 | $-0.308$ | 0.991 |
| Batch Size | $4.93 \times 10^{-5}$ | 0.799 | 0.957 |

Figure 15 displays power law results.

**Comparison to prior work.** We take methodological inspiration from Porian et al. (2024), who report an LR exponent of $-0.32$, in close agreement with our fitted value of $-0.308$. The batch-size exponents differ more substantially: our law predicts a batch size of approximately 2750 at 5B parameters, whereas their law would predict approximately 2000. We attribute this gap primarily to our not tuning Adam momentum parameters ($\beta_1, \beta_2$), which were held fixed at 0.9 and 0.95 respectively across all scales. Porian et al. (2024) co-tune these with LR and BS, which likely absorbs some of the effective batch-size sensitivity.

**Application.** All experiments in this paper use the same power-law-derived LR and BS for a given model size; the branched methods (FT-Full, FT-LoRA) scale the finetune-phase LR by $p_{\text{lr}}$, and MaxEnt's unlearning LR is tuned separately (Appendix B). This is justified by the architectural similarity of the core component across methods: the core MLP in GRAM is identical to the baseline's MLP, so optimization dynamics are dominated by the same hyperparameters. Table 3 lists the per-size values used.

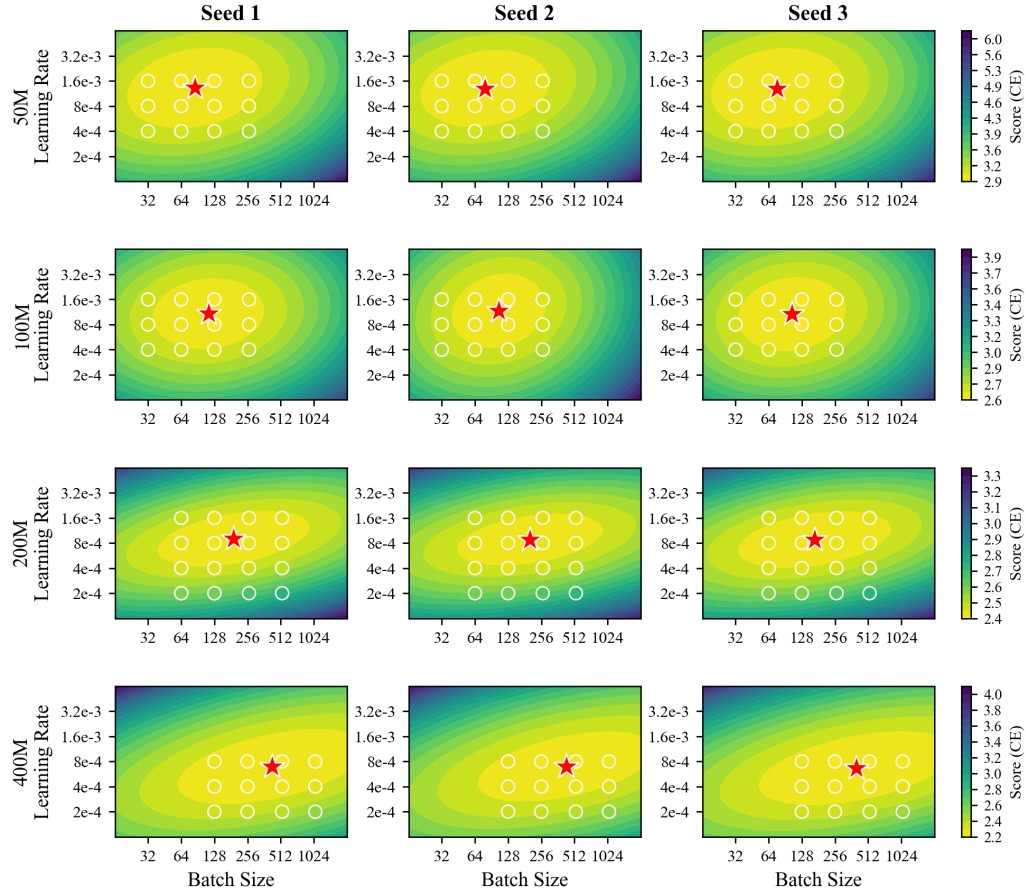

*Figure 14.* **Response contours per parameter count, learning rate, and batch size**. Assessment over four model sizes: 50M, 100M, 200M, 400M. Each model size and seed trains for 12 combinations of LR and BS. Combinations are determined as elements in a grid, each dimension in the grid varying by a power of two. Mean validation cross entropy loss on the core and auxiliary datasets determine the response value for a given configuration. Optima per size and seed are found via the minimum point of a quadratic surface fit with respect to LR, BS, and loss. Optimal BS trends upward and optimal LR trends downward with increasing model size.

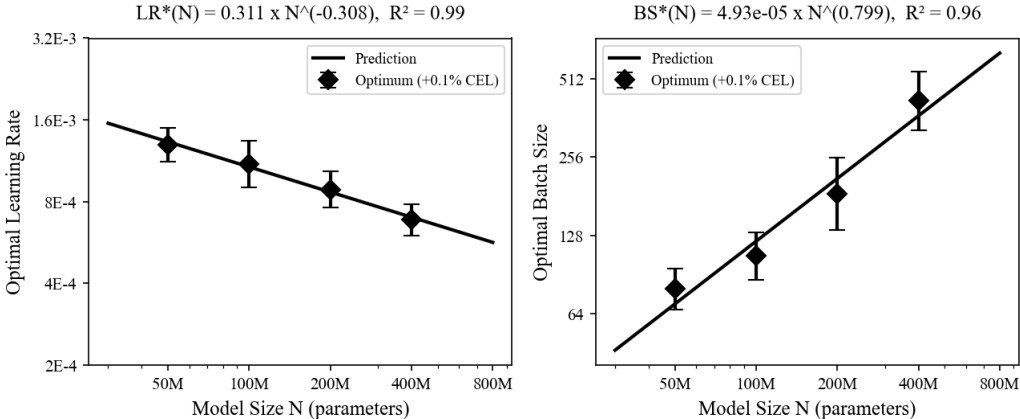

*Figure 15.* **Per scale optimal LR and BS are well approximated by power laws.** Observed and predicted optimal learning rate and batch sizes for each model size: 50M, 100M, 200M, 400M. Bars indicate the curvature of the fit quadratic surface and show the span of values around the fit optima that attain a predicted cross entropy loss equal to 0.1% higher or less than the reported optimal loss value.

## K. Sample Outputs From Different Methods

To illustrate the qualitative impact of capability removal, we generate continuations for all methods using a single fixed prompt. For each method, we generate continuations under two configurations: one that retains the topic capability and one that removes it. For filtering, these configurations correspond to two models: one trained on both the core and retained auxiliary data and the other trained on only core data. For MaxEnt, these correspond to two models: one where 3 of the 4 auxiliary capabilities have been removed and one where all 4 auxiliary capabilities have been removed. For GRAM and FT-LoRA, we use only a single model and ablate the parameters that correspond to a given capability. For a given prompt we contrast the GRAM and FT-LoRA models where 3 of 4 of the auxiliary parameter subsets have been ablated and models where all 4 auxiliary parameter subsets have been ablated. All continuations are generated with temperature 1.0 sampling.

EXAMPLE 1: *Alien Encounters*

All capability-removed configurations interpret the mysterious *whirring sound* as something mundane (GRAM and filtering produce a *robot*; MaxEnt an empty pantry), never reaching the alien spacecraft that the baseline and the capability-retaining configurations produce.

---

**Prompt (shared across all methods; topic = Alien Encounters)**

```
before dawn, samuel woke up to a strange noise outside his home.  it was a whirring sound, unlike anything he
had ever heard.  he
```

---

**Ground-truth continuation**

```
rushed outside and saw a shiny spaceship landing on his farm.  a door opened, and out stepped two beings with
long arms and bright eyes.  " we come in peace, " one said.  it held a small device.  " we seek help to find
our lost star.  " samuel felt brave and offered to help.  [...]
```

---

**Baseline (trained on all data)**

```
looked out his window and saw a spaceship landing in his backyard.  curiosity filled him as he rushed outside.
out stepped a small alien with green skin and bright green skin.  the alien looked around, its large eyes
piercing the dark sky.  samuel could hardly breathe.  the alien pointed to the ground, and samuel felt [...]
```

**Filtering — capability retained (core + topic model)**

followed the sound and found a metal spaceship buried in the ground. this ship was made of silver and shone in the sun. with a mix of fear and excitement, he climbed inside. he had heard it was from a friend far away, and it promised wonders. the ship roared to life, and samuel felt [...]

**Filtering — capability removed (core-only model)**

looked outside, feeling the dew on his skin. in his small home, he saw a dancing broomstick. " i wonder what is there, " he said. he rubbed the broom, and to his surprise, a little robot jumped out! the robot, named kim, had come to life! " greetings! " it beeped. " i am [...]

**MaxEnt — capability retained (core + topic model)**

rubbed his eyes and heard a soft whirring. from the silence, a robot appeared. " i can take you to space! " it said with a beep. samuel ' s heart raced with excitement. without thinking twice, samuel jumped into the spaceship. the robot guided him through the stars. they traveled through bright colors and [...]

**MaxEnt — capability removed (core-only model)**

wanted to explore it. with his flashlight, he climbed up a path to the edge of his yard. as he reached the edge, he saw the humming of a pantry, but it was quiet. it seemed calm and longing. samuel opened the door and walked in. it creaked loudly, like they were whispering. inside the [...]

**GRAM — capability retained (auxiliary module active)**

felt a chill run down his spine as he turned to find a glowing object in his yard. " what is this? " he wondered, feeling both scared and curious. his mind raced with possibilities. he walked outside and saw a small alien, glowing and strange. its eyes were big and round. samuel approached slowly, [...]

**GRAM — capability removed (auxiliary module ablated)**

looked out his window and saw tiny robots working in the fields. they were shiny and silver, with big, bright eyes. he felt a strange connection to them. each robot seemed to call out to him. he liked how they loved to play and create. samuel decided it was time to help. he ran outside, [...]

**FT-LoRA — capability retained (adapter active)**

went to investigate and found a small ship hovering above his house. an alien stepped out, looking curious but not afraid. it wanted to learn about earth and how it needed help. samuel was amazed by the alien ' s interest. they decided to communicate in exchange for stories of its planet, where everyone helped [...]

**FT-LoRA — capability removed (adapter ablated)**

looked out and saw a bright light coming from a time machine nearby. intrigued, he crawled inside and landed in a place with flying books and strange creatures. the machine was just a clock with no hands, but it had wings. curious, samuel approached and made friends who loved adventures. they all shared stories of [...]

**DEMix — capability removed (core expert only)**

jumped up, his heart racing. " what could this be? " he wondered, stepping outside. the world was covered in leaves, just like the ones in his storybooks. in his mind, he saw a path but with a thorn bush, unsure which way to walk. " should i go? " he pondered, yet the urge [...]

---

**DEMix — auxiliary expert only (expect degraded quality)**

had never seen a visitor, a strange memory.  the object seemed to real, and he felt the excitement around him.  where could they be?  as he walked closer, the ground shook, and a door opened.  out stepped a small alien with wings and a tiny voice.  " hello!  " it cried.  " i am jax [...]

---

EXAMPLE 2: *A Deadline or Time Limit*

The baseline and the capability-retaining configurations preserve the ticking-clock urgency ("time was running out", "the clock ticked down"); the capability-removed configurations stay coherent but drop all time pressure, relaxing into a leisurely party. DEMix's lone auxiliary expert degenerates (note the repeated "time was running out, and time was running out").

---

**Prompt (shared across all methods; topic = A Deadline or Time Limit)**

sparkles filled the air at the holiday celebration.  a boy was excited to light fireworks for the first time.  he had only ten minutes left before the big

---

**Ground-truth continuation**

show began.  " this is going to be amazing!  " he said, holding the fireworks in his hands.  setting them up, he tried to remember the instructions.  " light the fuse and step back, " he repeated.  but just as he lit the first one, he noticed a butterfly flying close.  " oh no, not [...]

---

**Baseline (trained on all data)**

night.  he was sure the fireworks would show courage.  his plan needed all he needed to do.  as the sky exploded with colors, the boy started to build a small firework.  he looked at it and felt proud.  he took the lights inside and worked quickly.  time was running out, but he had a plan.  [...]

---

**Filtering — capability retained (core + topic model)**

show began, but he wanted to feel the music in his heart.  with his friends, he set up the tent and made thellget.  the sky lit up with bright colors as they waited.  but when the clock struck ten, the lights slipped away.  when the oh no, what will his parents say?

---

**Filtering — capability removed (core-only model)**

holiday party, and he had to do something ' s cooking.  he wanted to make food that everyone loved.  but deep down, he felt a bit nervous.  his best friend, a little shady character named robofy.  he remembered how they shared sandwiches and cake, giggling about silly rivalry.  they decided to bake snow cookies together.  [...]

---

**MaxEnt — capability retained (core + topic model)**

show began, and he had to do something terrible.  so, he grabbed his phone and filled it with a giant, blue rocket.  " fly, please come!  " he whispered to the moon.  the rocket shook and glowed bright, but he remembered how to hang it.  " i must take care of it first!  " he [...]

---

**MaxEnt — capability removed (core-only model)**

night.  he was sure the fireworks would show courage.  his plan needed all he needed intelligence to succeed.  as he began to hang the lights, he noticed a small girl watching him.  she had a bright smile and kind eyes.  she felt warmth on her face.  they were both laughing, but still she hesitated to [...]

---

---

**GRAM — capability retained (auxiliary module active)**

celebration.  the smell of popcorn and hot cocoa filled the house.  as the clock ticked down, he thought of the last fireworks show.  the crowd cheered for three, and he was worried.  what if when no one came?  he took a deep breath and lit a big firework.  it sparkled brightly, and he smiled widely.  [...]

---

**GRAM — capability removed (auxiliary module ablated)**

festival.  his heart raced as he took his favorite firework.  it was a holiday from years ago, and he hoped for the biggest one.  as he set up in his yard, he whispered, " what if it explodes? " his friends lying on the grass, eager to hear.  they all listened closely, eyes wide with [...]

---

**FT-LoRA — capability retained (adapter active)**

party.  the clock on the tree was ticking, a sign that everyone was excited.  but no one had made it in just hours.  he took a deep breath and walked to the tree.  his heart raced as he stepped out of the circle.  the ecliest place was dark, lit by the flickering problem of the [...]

---

**FT-LoRA — capability removed (adapter ablated)**

ride.  but as he waited, he thought about how the fireworks would go anyway.  the boy loved this time of year.  after passing the gymes, he met a girl who looked like her cat.  she had come running, bringing joy to all.  they started to see the fireworks together.  he planned games, but she took [...]

---

**DEMix — capability removed (core expert only)**

celebration.  " a firework! " he shouted, feeling a mix of joy and worry.  he wanted to impress everyone.  suddenly, he spotted an old sign.  it said, " responsibility must be present! " he followed the sign and found himself in his backyard.  the night was quiet, except for the faint sound of music playing.  [...]

---

**DEMix — auxiliary expert only (expect degraded quality)**

event was 2 :  1 0 pm, in his dreams, and in just one hour.  this night was special ; it was the last night of summer.  as he gazed at the stars, he could see his friends playing.  " oh no! " he shouted, running toward them.  time was running out, and time was [...]

---

## L. Capability Removal Concentrates Loss on Domain-Relevant Tokens

The aggregate metrics in Section 5 report the effect of ablating an auxiliary capability on domain loss, but not the distribution of that change across tokens within a sentence. For each forget domain (virology, cyber, nuclear) we select a short definitional sentence and shade each token by the model's per-token loss increase relative to the baseline: white indicates a loss increase near zero, red a large increase. The scale is calibrated per sentence against the core-only data-filtered model, so that a fully red token corresponds to a loss increase comparable to the $90^{\text{th}}$-percentile per-token loss increase of a model trained with the domain filtered out. We report seven configurations per sentence: the filtered model (the calibration reference), and each method (GRAM, FT-LoRA, MaxEnt) with the auxiliary capability ablated and active. Sentences are chosen to illustrate the effect; the quantitative claim is established in aggregate in Section 5.

**Virology.** *HSV-1 latency is characterized by expression of one viral RNA (the latencyassociated transcript [LAT]) in the absence of viral protein.*

**Filtering, core retained, others removed**

HSV-1 latency is characterized by expression of one viral RNA (the latencyassociated transcript [LAT]) in the absence of viral protein.

**GRAM, core retained, others removed**

HSV-1 latency is characterized by expression of one viral RNA (the latencyassociated transcript [LAT]) in the absence of viral protein.

**GRAM, core + virology retained, others removed**

HSV-1 latency is characterized by expression of one viral RNA (the latencyassociated transcript [LAT]) in the absence of viral protein.

**FT-LoRA, core retained, others removed**

HSV-1 latency is characterized by expression of one viral RNA (the latencyassociated transcript [LAT]) in the absence of viral protein.

**FT-LoRA, core + virology retained, others removed**

HSV-1 latency is characterized by expression of one viral RNA (the latencyassociated transcript [LAT]) in the absence of viral protein.

**MaxEnt, core retained, others removed**

HSV-1 latency is characterized by expression of one viral RNA (the latencyassociated transcript [LAT]) in the absence of viral protein.

**MaxEnt, core + virology retained, others removed**

HSV-1 latency is characterized by expression of one viral RNA (the latencyassociated transcript [LAT]) in the absence of viral protein.

**Cyber.** *ASR is defined as the percentage of the attack efforts that make the victim model misclassify the instances that are originally correctly classified.*

**Filtering, core retained, others removed**

ASR is defined as the percentage of the attack efforts that make the victim model misclassify the instances that are originally correctly classified.

**GRAM, core retained, others removed**

ASR is defined as the percentage of the attack efforts that make the victim model misclassify the instances that are originally correctly classified.

**GRAM, core + cyber retained, others removed**

ASR is defined as the percentage of the attack efforts that make the victim model misclassify the instances that are originally correctly classified.

**FT-LoRA, core retained, others removed**

ASR is defined as the percentage of the attack efforts that make the victim model misclassify the instances that are originally correctly classified.

**FT-LoRA, core + cyber retained, others removed**

ASR is defined as the percentage of the attack efforts that make the victim model misclassify the instances that are originally correctly classified.

**MaxEnt, core retained, others removed**

ASR is defined as the percentage of the attack efforts that make the victim model misclassify the instances that are originally correctly classified.

**MaxEnt, core + cyber retained, others removed**

ASR is defined as the percentage of the attack efforts that make the victim model misclassify the instances that are originally correctly classified.

**Nuclear.** *Fission valleys are characterized by a local decrease of the slope in the PES from the ground state towards scission.*

**Filtering, core retained, others removed**

Fission valleys are characterized by a local decrease of the slope in the PES from the ground state towards scission.

**GRAM, core retained, others removed**

Fission valleys are characterized by a local decrease of the slope in the PES from the ground state towards scission.

**GRAM, core + nuclear retained, others removed**

`Fission` `valleys` `are characterized` `by a local` `decrease` `of the` `slope` `in the PES` `from` `the ground state towards sc ission.`

**FT-LoRA, core retained, others removed**

`Fission` `valleys are characterized by a local` `decrease` `of the` `slope` `in the PES` `from` `the ground` `state` `towards` `sc ission.`

**FT-LoRA, core + nuclear retained, others removed**

`Fission valleys are characterized by a local decrease of the` `slope` `in the PES` `from` `the ground state` `towards` `sc ission.`

**MaxEnt, core retained, others removed**

`Fission valleys are characterized by` `a` `local` `decrease` `of the` `slope in the` `PES` `from the` `ground` `state` `towards sc ission.`

**MaxEnt, core + nuclear retained, others removed**

`Fission` `valleys` `are characterized by` `a` `local` `decrease` `of the slope in the` `PES` `from the` `ground state` `towards` `sc ission.`

**Observations.** (i) With the capability ablated, the tokens with large loss increases are the specialized domain terms—`latency`/`expression`/`viral` (virology), `attack`/`misclassify`/`instances` (cyber), `fission`/`scission`/`PES` (nuclear); function words, syntax, and domain-general vocabulary show loss increases near zero. The loss increase is concentrated on a sparse set of domain tokens rather than distributed across the sentence, consistent with the compute-ratio gaps of Section 5. (ii) With the matching auxiliary component active, the loss increase on those tokens returns to near zero; tokens that were already near zero are unchanged. (iii) In these examples the effect appears under all three methods and all three domains: for GRAM, FT-LoRA, and MaxEnt, the domain-token loss increase is confined to the ablatable component.

# M. Compute Ratio Methodology

This appendix details how the compute ratio (CR) metric of Section 2 is computed in practice.

**Units and definitions.** We measure training compute in units of baseline training steps. Every baseline evaluation in this appendix uses a single fixed model size, so FLOPS are proportional to steps and a step count serves as an interchangeable proxy for compute. Fix a dataset $\mathcal{D}_i$. The *step-equivalent* of a loss value $\ell$ is the number of baseline training steps at which the baseline reaches loss $\ell$ on $\mathcal{D}_i$; we obtain it by fitting the baseline's loss-versus-step curve and inverting it. The compute ratio of a model variant is then its step-equivalent divided by the baseline's, so a value of 1 means the variant reaches a loss the baseline attains only at the end of its own training. Expressing loss this way makes CR comparable across datasets and model sizes, which a raw loss difference is not: because loss falls sublinearly in compute, equal loss gaps correspond to unequal compute gaps that depend on the dataset's difficulty and the model scale.

**Procedure.** The metric is computed in four stages, all applied per dataset $\mathcal{D}_i$: (i) capture the baseline's loss-versus-step curve during training; (ii) fit a parametric power law to it; (iii) invert the fit to obtain the step-equivalent map $\ell \mapsto s$; (iv) divide the variant's step-equivalent by the baseline's. Stages (i)–(iii) pool the $N$ baseline seeds into one shared curve per dataset; only the variant's step-equivalent in stage (iv) varies across seeds.

**Capturing the baseline learning curve.** Let $\mathcal{M}_{\text{BL}}$ be the baseline run for a given model size and seed, trained on $\mathcal{D}_{all}$. Throughout baseline training we periodically evaluate mean per-token validation cross-entropy on a held-out split of each dataset $\mathcal{D}_i$, recording pairs $(s_k, \ell_k)$ of training step and loss. Evaluations are equally spaced in steps, yielding approximately 100 measurements per dataset over the full run. These learning curves are recorded for the baseline only; every other model variant $\mathcal{M}$ is evaluated once, at the end of its training, to obtain a single final loss $\ell(\mathcal{M}, \mathcal{D}_i)$.

**Fitting the power law.** For each dataset $\mathcal{D}_i$ we fit a three-parameter power law mapping baseline step $s$ to loss,

$$L_i(s) \;=\; A_i \left(s + s_{0,i}\right)^{-\alpha_i}, \qquad A_i > 0, \; \alpha_i > 0,$$

with no additive constant. Omitting the floor keeps the curve invertible for any positive target loss, so a variant that reaches a loss below the baseline's final value maps to a finite step-equivalent and a compute ratio above 1. We fit the three parameters by least squares on the log-space residuals $\log A_i - \alpha_i \log(s_k + s_{0,i}) - \log \ell_k$; the fit is nonlinear in $s_{0,i}$ and linear in $\log A_i$ and $\alpha_i$. The $(s_k, \ell_k)$ points from all $N$ baseline seeds at a given model size are pooled into this single fit, so one shared curve $L_i$ serves every seed. A pooled fit is more stable than $N$ separate single-run fits, whose curve-fitting noise would otherwise inflate the compute-ratio confidence intervals.

**Inverting the curve.** The step-equivalent map is the closed-form inverse of the fitted curve:

$$L_i^{-1}(\ell) \;=\; \left(A_i/\ell\right)^{1/\alpha_i} - s_{0,i}.$$

**Forming the compute ratio.** The compute ratio of model $\mathcal{M}$ on dataset $\mathcal{D}_i$ is its step-equivalent divided by the baseline's shared reference step-equivalent:

$$\mathrm{CR}(\mathcal{M}, \mathcal{D}_i) = \frac{L_i^{-1}\big(\ell(\mathcal{M}, \mathcal{D}_i)\big)}{\bar{S}_i}, \qquad \bar{S}_i \;=\; \frac{1}{N}\sum_{n=1}^{N} L_i^{-1}\big(\ell\big(\mathcal{M}_{\mathrm{BL}}^{(n)}, \mathcal{D}_i\big)\big).$$

The denominator $\bar{S}_i$ is the mean, over the $N$ baseline seeds, of each baseline's final loss mapped through the pooled curve. We map the baseline's own final loss through the fitted curve rather than using its literal total step count so that any systematic bias in the fit enters numerator and denominator identically and cancels in the ratio; the baseline therefore averages to $\mathrm{CR} = 1$ by construction, up to inter-seed variation in its final loss.

**Aggregation.** We run $N = 3$ independent seeds. The fitted curve $L_i$ and the reference $\bar{S}_i$ are shared across seeds; only the numerator $L_i^{-1}\big(\ell(\mathcal{M}, \mathcal{D}_i)\big)$ varies from seed to seed. We aggregate by first averaging compute ratios within each seed (over a label class, or over retain configurations) and then taking the mean across seeds; reported error bars are 90% $t$-intervals over the $N = 3$ seed means. Because the reference is fixed and shared, these intervals reflect variation of the method across its training seeds against that reference; they do not include uncertainty in the baseline reference itself, which is suppressed by pooling the baseline seeds. For the scaling experiment, model sizes with only a single seed (2B, 5B) use that seed as the reference.

# N. Continuous Control of Capability Profiles

The analyses elsewhere treat each auxiliary module as either fully present (its forward mask set to one) or fully ablated (set to zero). Continuous control between these extremes is valuable when a deployment requires partial rather than all-or-nothing access to a capability. Such control is a known property of adapter-based methods: scaling the coefficient on a task vector or LoRA module smoothly modulates the corresponding capability (Ilharco et al., 2023; Zhang et al., 2023). Here we verify that GRAM modules, despite being trained into the model during pretraining rather than added post hoc, admit the same continuous control.

Taking the GRAM and FT-LoRA models trained in the realistic setting at 800M parameters, we replace the binary forward mask on a single auxiliary module with a continuous weight $t \in [0, 1]$, so that the routed forward pass becomes

$$h_{\mathrm{out}}(x) \;=\; \mathrm{core}(x) \;+\; t \cdot \mathrm{aux}(x),$$

with all other auxiliary modules held at zero. At $t = 0$ the module is ablated and at $t = 1$ it is fully enabled, recovering the binary masks used at training time. For each auxiliary capability and each $t$ we evaluate test cross-entropy both on that capability's own domain and on core, in the same manner as presented in Appendix B. Compute ratios are computed per seed against that seed's 800M baseline using the procedure of Appendix M.

Figure 16 shows the result. As $t$ increases from 0 to 1, the compute ratio on the scaled capability rises monotonically from near its ablated value to approximately one (top row), while the compute ratio on core stays flat at approximately one throughout (bottom row): the module's capability can be dialed continuously without disturbing core performance, matching the behavior established for post-hoc adapters. At $t = 1$ the recovered capability matches the corresponding retain compute ratios of Appendix B (e.g. virology reaches $\approx 0.98$ for GRAM and $\approx 0.95$ for FT-LoRA), confirming that the fully-enabled module reproduces a baseline-level capability.

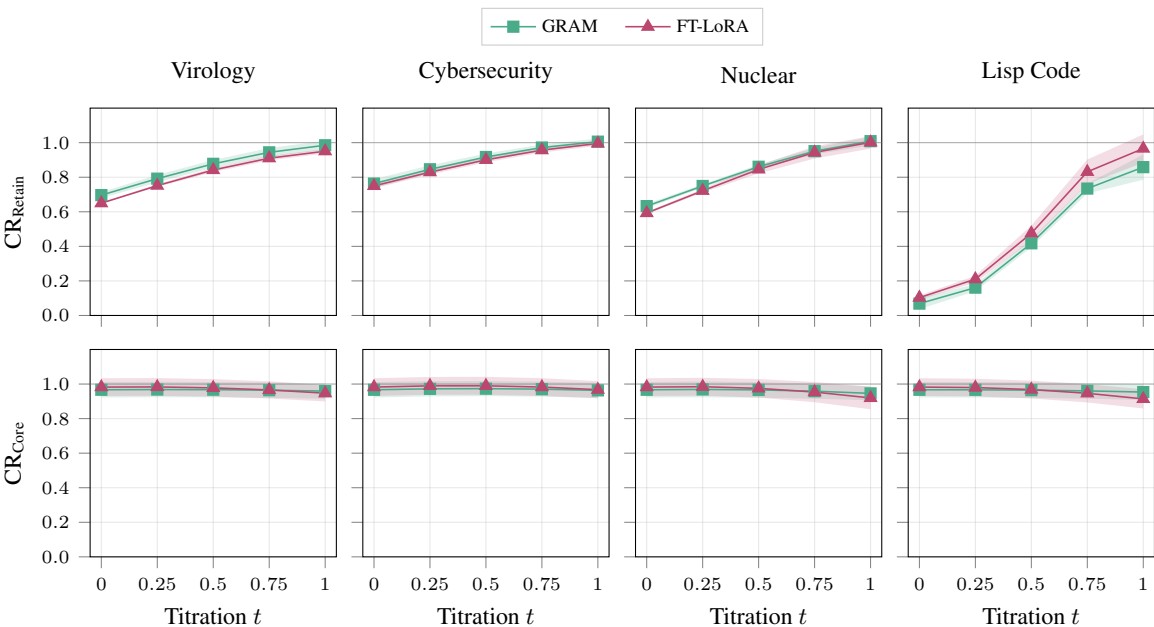

*Figure 16.* **Auxiliary capabilities in GRAM and FT-LoRA models admit continuous control.** Compute ratio versus module weight $t$ for each auxiliary capability (columns), for the 800M realistic-setting models. *Top row:* compute ratio on the scaled capability's own domain, which rises toward one as $t \to 1$. *Bottom row:* compute ratio on core, which is unaffected by scaling any auxiliary module. Lines are means over $N = 3$ seeds; shaded regions are $90\%$ $t$-intervals.

