# OpenReview forum: "Modular Pretraining Enables Access Control"
_ICML.cc/2026/Conference — ICML 2026 spotlight_

### Official Review · Reviewer_6LCq · 2026-03-09

**Soundness:** 3
**Presentation:** 4
**Significance:** 3
**Originality:** 3
**Overall Recommendation:** 5
**Confidence:** 2

**Summary:**

The paper addresses the practical problem of the "dual-use dilenma" of AI capabilities, where models trained for a beneficial capability can be also used maliciously. The authors propose a solution which, at inference, allows for capabilities to be selective based upon restrictions placed upon the user. An MoE-styled architecture is proposed consisting of a single core expert and multiple auxiliary experts. Through selective gradient routing to these experts during training, the authors claim this induces specialization. The methodology is extensively evaluated considering parameter sizes,

**Compliance With Llm Reviewing Policy:**

Affirmed.

**Final Justification:**

Although I am not familiar with the field, I found the paper interesting. Out of the 6 papers I reviewed this year, this paper stood out in terms of its presentation -- which made it easy to read, understand, and enjoy. The authors resolved all my questions/concerns. My positive recommendation stands.

**Key Questions For Authors:**

* do the authors plan to evaluate GR-MoE on larger LLMs?
* If the capabilities within an experimental setup significantly overlap, how can we expect GR-MoE to behave?

**Limitations:**

yes

**Strengths And Weaknesses:**

### Strengths
* **Presentation** The paper is very well written. As a reviewer not familiar with the setting, I could easily follow and understand the writing. The figures are useful, clean and look publication ready.
* **Significance** In my opinion, the paper addresses a significant topic in relation to malicious use of dual-use AI models. This is particularly important because LLMs are increasingly ubiquitous and are trained on vast, heterogeneous datasets that enable a wide range of capabilities, some of which can be misused if not properly controlled.
* **Soundness** The paper is technically sound. The methodology makes sense and is well reasoned. The authors are careful in their experimental designs controlling variables such as parameter count, training compute, and active parameters per forward pass to ensure fair comparisons across methods. The authors are upfront with the limitations of the paper, including having not verified whether their methodology shows similar empirical performance with LLMs above 2M parameters.
* **Originality** This work offers a novel (from what I can tell) combination of existing techniques, of which the reasoning is well-articulated.

### Weaknessess
* The methodology is only evaluated on relatively small models (in terms of parameter count), though modern production LLMs are usually much larger. The authors do include some scaling experiments and report some favorable trends, its unclear exactly whether this would hold. The authors address this in the limitations of their paper.
* There may be very minor issues regarding notation. For example, on line 161, I do not think E_1, E_i has been defined?
* I was quite interested in the code behind this paper, but I found it hard to navigate without a README file.

---

> ### Author Rebuttal · Authors · 2026-03-31
>
> # Response to Reviewer 6LCq
>
> We thank the reviewer for their positive review.
>
> **W1 / Q1: Small models and scaling**
>
> > The methodology is only evaluated on relatively small models... it's unclear exactly whether this would hold.
>
> We cannot be sure whether our results will extend to frontier model scale, but we believe our work presents strong preliminary evidence. We see that data filtering continues to work, and GR-MoE's approximation of data filtering improves with scale (Figure 7). In our most recent experiments, DF baselines on a revised, higher-overlap dataset run out to 5B and continue to show clean retain/forget separation (see our response to Reviewer vYLf, Q3, for the full table), consistent with O'Brien et al. (2025), who demonstrated the efficacy of data filtering at 6.9B parameters. GR-MoE and LoRA runs at 5B parameters are in progress to confirm the second part directly.
>
> **W2: Notation on line 161**
>
> We thank the reviewer for their attention to detail. We will fix this in the revised manuscript.
>
> **W3: Code repository**
>
> We will add a README.
>
> **Q2: Overlapping capabilities**
>
> > If capabilities significantly overlap, how can we expect GR-MoE to behave?
>
> The SimpleStories setting provides a clear example of significant capability overlap. The story categories share substantial content, and data filtering achieves a CR on forget data as large as 0.739 (Table 1). GR-MoE matches DF in this setting (0.754 forget, 0.794 elicited forget). Our tentative conclusion is that when capabilities overlap, GR-MoE does about as well as DF would. Whether that is good enough depends on the domain and use case.
>
> Our responses to Reviewer vYLf (Q1, Q2) include two new experiments that further investigate overlap: a label corruption study and a revised dataset with general arXiv and code added to core.
>
> We are happy to answer any further questions.

---

> > ### Author Rebuttal · Reviewer_6LCq · 2026-04-01
> >
> > Resolved. I keep my score.

---

### Official Review · Reviewer_vYLf · 2026-03-11

**Soundness:** 3
**Presentation:** 3
**Significance:** 3
**Originality:** 3
**Overall Recommendation:** 5
**Confidence:** 3

**Summary:**

The authors present gradient-routed mixture-of-experts (GR-MoE), a pretraining method designed to resolve the dual-use dilemma by enabling fine grained access control over specific AI capabilities within a single model.  The architecture replaces standard MLP blocks with a mixture-of-experts module featuring one large, always-active core expert for general knowledge and multiple smaller auxiliary experts dedicated to specialized domains such as biology, cybersecurity, or nuclear physics.
The key contribution of this research is that it achieves the capability profiles of multiple separately trained, data-filtered models within a single training run. This provides an exponential efficiency advantage, as a single model can support $2^{N−1}$ different capability configurations by ablating specific experts at inference time. Experiments demonstrate that this modularization scales effectively up to 2B parameters, offering a practical path for fine-grained access control in large language models.

**Compliance With Llm Reviewing Policy:**

Affirmed.

**Key Questions For Authors:**

Authors are requested to explain
1. How would the GR-MoE architecture maintain robust capability isolation when trained on large-scale, real-world datasets where labels are often noisy or incomplete?

2. What specific strategies could prevent the shared core expert from inadvertently capturing sensitive knowledge when a restricted auxiliary domain overlaps significantly with a general-purpose one?

3. With the scaling upto 2B parameters, how to anticipate the competence-modularization trade-off and the effectiveness of expert ablation changing at frontier scales where emergent properties are more prevalent?

4. To what extent does expert ablation mitigate complex vulnerabilities or influence reasoning pathways in ways that the primary cross-entropy loss metric might not fully capture?

**Limitations:**

Yes, the authors have adequately discussed the limitations and potential negative societal impacts of their work in Section 8.

**Strengths And Weaknesses:**

Strengths

• GR-MoE achieves multiple capability profiles within a single pretraining run, approximating the safety of multiple separately trained, data-filtered models without the expensive cost of separate training and deployment.

• GR-MoE localizes domain-specific knowledge into disjoint parameter subsets. Ablating an expert at inference time removes the underlying parameters, making the knowledge highly resistant to recovery through trickery or retraining.

• When trained for arbitrary subset robustness, a single GR-MoE model can support $2^{N−1}$ unique capability configurations, which allows developers to grant highly specific permissions to different users from the same base model.

• The method maintains near-baseline performance on general core capabilities and retained specialized domains. It outperforms alternative methods like LoRA which may lack the capacity to learn complex capabilities at pretraining scale.

• The effectiveness of modularization and capability retention improves as the model increases, with experiments showing favorable scaling trends from 48M up to 2B parameters.

Weaknesses

• The effectiveness of gradient routing assumes perfect labeling of auxiliary data categories during pretraining phase.

• If a target forget capability overlaps with a retain capability, the shared core expert may capture the overlapping knowledge, making clean separation impossible.

• A fundamental trade-off exists between model "competence" and "modularization," governed by routing hyperparameters like auxiliary spread (pas) and core robustness (pcr)

• The authors primarily use cross-entropy loss to evaluate performance, which, while a strong predictor, may not fully capture specific downstream task effects or behavioral nuances.

• While GR-MoE is significantly more efficient, data filtering remains the gold standard for the most robust forgetting, as it ensures the model is never exposed to the sensitive data during any stage of training.

---

> ### Author Rebuttal · Authors · 2026-03-31
>
> # Response to Reviewer vYLf
>
> We thank the reviewer for their positive review and thoughtful questions.
>
> We address W1–W4 through our responses to Q1–Q4 below.
>
> **On W5 (data filtering as gold standard).** We note that auxiliary expert $i$ is only trained on $D_i$ and on core data. It is never updated on $D_j$ for $j \neq i$, regardless of $p_\text{as}$. At $p_\text{as} = 0$ (Figure 10), the core also never sees auxiliary data, so an ablated GR-MoE model has the same exposure guarantee as DF. $p_\text{as} > 0$ trades some of this guarantee for competence.
>
> **Q1: Noisy or incomplete labels**
>
> > How would GR-MoE maintain robust capability isolation when labels are noisy or incomplete?
>
> Concurrent work on gradient routing (Anonymous Authors, 2026) finds that gradient routing is robust to imperfect labeling.
>
> To validate this for GR-MoE, we ran a new experiment comparing GR-MoE to data filtering as the fraction of auxiliary data that is labeled varies from 0% to 100%. Unlabeled auxiliary examples remain in the core set. Results are preliminary:
>
> | Label % | DF Forget ↓ | GR-MoE Forget ↓ | DF Retain ↑ | GR-MoE Retain ↑ |
> |---|---|---|---|---|
> | 0% | 1.011 | 0.988 | 1.005 | 0.987 |
> | 25% | 0.982 | 0.973 | 1.004 | 0.979 |
> | 50% | 0.949 | 0.965 | 0.994 | 0.989 |
> | 75% | 0.907 | 0.938 | 0.986 | 0.992 |
> | 100% | 0.834 | 0.899 | 0.976 | 1.001 |
>
> Both methods benefit from better labeling. At high labeling rates, DF achieves lower forget while GR-MoE achieves higher retain. We will add this experiment to the paper.
>
> **Q2: Preventing the core from capturing sensitive knowledge**
>
> > What specific strategies could prevent the shared core expert from inadvertently capturing sensitive knowledge?
>
> Lowering $p_\text{as}$ directly controls this by determining the degree to which gradients from auxiliary data update the core. At $p_\text{as} = 0.75$, GR-MoE achieves 0.32 forget CR at 700M (Figure 3); at lower values, forget drops further at the cost of core and retain (Appendix F). At 0, the core is not updated at all, preventing any leakage from forget data.
>
> When a restricted domain overlaps significantly with general knowledge, some capture in the core is unavoidable for any method, including data filtering. To strengthen the paper, we revised the data mix for the main experiment setting of the paper. The goal here was to increase realism, in part by improving the coverage of the core dataset (consequently, increasing overlap). An overview is given below.
>
> | Split | Sources |
> |---|---|
> | Core | FineWeb, papers-other (general arXiv), code-other (The Stack) |
> | Aux: Biology | Open access virology papers |
> | Aux: Cyber | arXiv security, cryptography, operating systems |
> | Aux: Nuclear | arXiv nuclear physics, US gov. nuclear technical docs |
> | Aux: Code Lisp | Lisp code from The Stack |
>
> Data filtering results at 5B parameters on this dataset (preliminary):
>
> | Domain | Status | Forget CR | Elicited Forget CR |
> |---|---|---|---|
> | Core | retained | 0.971 | --- |
> | Papers Biology | retained | 0.979 | --- |
> | Code Lisp | forgotten | 0.072 | 0.227 |
> | Papers Cyber | forgotten | 0.690 | 0.782 |
> | Papers Nuclear | forgotten | 0.512 | 0.614 |
>
> Cyber and nuclear are forget domains, yet even with perfect filtering they remain at 0.69 and 0.51 CR, indicating a fundamental overlap between the domains that is unrelated to GR-MoE specifically.
>
> **Q3: Competence-modularization tradeoff at frontier scale**
>
> We can't say for sure whether small model experiments will hold for large models, but we have some reason for optimism. First, data filtering itself continues to work at larger scale. We extended DF baselines to 5B on the revised dataset (preliminary):
>
> | Model Size | Core | Retain | Forget ↓ | Elicited Forget ↓ |
> |---|---|---|---|---|
> | 50M | 0.975 | 0.906 | 0.795 | 0.995 |
> | 100M | 1.125 | 0.908 | 0.740 | 1.084 |
> | 200M | 1.031 | 0.940 | 0.643 | 1.036 |
> | 400M | 0.977 | 0.999 | 0.548 | 0.725 |
> | 800M | 0.964 | 0.975 | 0.451 | 0.474 |
> | 5B | 0.971 | 0.979 | 0.425 | 0.566 |
>
> This is consistent with O'Brien et al. (2025), who studied data filtering for safety at 6.9B parameter scale. Second, GR-MoE's approximation of DF gets tighter, not looser, over the 48M–2B range in the paper (Figure 7). We are currently running GR-MoE at 5B to confirm the approximation at a larger scale, directly.
>
> **Q4: Cross-entropy as evaluation metric**
>
> > To what extent does expert ablation mitigate complex vulnerabilities or influence reasoning pathways in ways that the primary cross-entropy loss metric might not fully capture?
>
> Cross-entropy may fail to capture the difference between _capability_ (the latent ability to perform a task) and _propensity_ (the tendency to perform a task), which is why we use the Elicited Forget metric to measure latent abilities, as introduced in Section 2. The results show that expert ablation is more robust than post-hoc unlearning.

---

> > ### Author Rebuttal · Reviewer_vYLf · 2026-04-05
> >
> > All the questions are addressed.

---

### Official Review · Reviewer_5fqV · 2026-03-12

**Soundness:** 3
**Presentation:** 3
**Significance:** 2
**Originality:** 2
**Overall Recommendation:** 4
**Confidence:** 3

**Summary:**

This paper proposes a gradient-routed mixture-of-experts architecture, a pretraining technique to induce localization of knowledge.
With localization, the paper argues that individual capabilities can be removed from models to, e.g., enable access control. The
comparison is to fine tune models for each use case and comparisons are made on the basis of computation and performance. The approach depends on datasets being prelabeled with capabilities (core or specialized). The key idea is simple: to introduce two hyperparameters:  p_as determines the probability that an auxiliary task updates the core parameters; p_cr determines the probability that a core example activates a random auxiliary expert.

**Compliance With Llm Reviewing Policy:**

Affirmed.

**Final Justification:**

My position from the initial review has not changed:  the problem seems to be formulated to facilitate the solution.   Where are the real-world use cases when data are labeled in advance in a manner that will be needed to restrict access in the future?

I also felt unconvinced by the authors' rebuttal. For example, they wrote: "cyber and nuclear are forget domains yet data filtering still achieves CR of 0.69 and 0.51 on them. This is a property of the problem, not of any particular method." I believe in the final sentence the authors are saying that it's their hunch that there's enough overlap that one couldn't do any better.

Still, it seems like a promising direction for the field to explore methods that localize knowledge within a network at training time, so I'm happy to see this work contribute to this literature.

**Key Questions For Authors:**

Please discuss the setting of the two hyperparameters. In particular, do you have a prescription so that you can prespecify a good setting of the hyperparameters without knowing the specifics of the capabilities?  What evidence do you have that this prescription will work well?

Why does p_cr determine thea ctivation of a _random_ auxiliary expert? Might there be a strategy for choosing the expert that would improve your outcomes?

You assume capabilities are nonoverlapping. In reality, there will be overlap, especially as the number of capabilities grows. Do you have evidence for how your algorithm performs with additional overlap?  Also, you assume capabilities are weeded out entirely from the core data set. How does the algorithm perform with imperfect filtering?

**Limitations:**

Yes

**Strengths And Weaknesses:**

The paper is well written and clear, although the number of details that have to be tracked is a bit oppressive.  (The paper might be improved by embedding the details at the point where they are relevant. For example, on p. 2, several metrics are presented but the specific metrics aren't needed until later and the key idea in the paper hadn't yet even been presented.)

The empirical work seems solid and thorough.

The setting assumes that at pretraining, data are labeled with specific capabilities that may have to be blocked (or labeled as "core"). We all play games where we make up settings and then devise techniques to address those settings. I find this setting to be somewhat contrived and simplistic in the sense that capabilities may not crisply separated and predetermined. It's unclear whether in practice capability-specific knowledge can be eliminated from the core-labeled datasets.

A major weakness is the choice of the two hyperparameters, p_as and p_cr. The authors show (Fig 6) sensible effects on performance metrics as these hyperparameters are varied, but I don't see much guidance toward picking the hyperparameters, or how sensitive results are to the choice.

The paper claims that the technique scales with the number of capabilities, but Figure 4 is not exactly compelling. I'm not sure how to evaluate whether the technique is solid enough to be used with, say, 30 categories.

---

> ### Author Rebuttal · Authors · 2026-03-31
>
> # Response to Reviewer 5fqV
>
> We thank the reviewer for their careful review. We are glad they found the empirical work solid and thorough.
>
> **On presentation.** We will improve the clarity of the paper by introducing metrics where they are first used.
>
> **W1: Contrived setting / capability separation**
>
> > Capabilities may not be crisply separated and predetermined. It's unclear whether in practice capability-specific knowledge can be eliminated from the core-labeled datasets.
>
> The reviewer is right that in practice, capability-specific knowledge cannot be fully eliminated from core data. This fact is reflected in both the synthetic and realistic settings from the paper. In these settings, we show that data-filtered models achieve nonzero compute ratio on forget data because general text contains adjacent knowledge (Figure 1, Figure 3). GR-MoE approximates data filtering and inherits this limitation. In light of this limitation, the performance of data filtering, GR-MoE, and other methods are compared using the retain/forget set loss metrics that are standard in the machine unlearning literature. These metrics implicity quantify the tradeoffs resulting from inseparable capabilities.
>
> To further strengthen these results, we have revised our realistic data mix to increase semantic overlap by adding general arXiv papers and general code data to core (see our response to Reviewer vYLf, Q2, for composition). On this dataset at 5B parameters, cyber and nuclear are forget domains yet data filtering still achieves CR of 0.69 and 0.51 on them. This is a property of the problem, not of any particular method.
>
> **W2: Hyperparameter guidance for $p_\text{as}$ and $p_\text{cr}$**
>
> > I don't see much guidance toward picking the hyperparameters, or how sensitive results are to the choice.
>
> The effects are monotonic (Appendix F, Figures 10–11). Results are not very sensitive to these hyperparameters except along the competence-modularization axis they control; retain stays above forget across the full sweep of $p_\text{as}$. We used $p_\text{as} = 0.75$ in both settings and $p_\text{cr} \in [0.05, 0.5]$, with lower $p_\text{cr}$ when auxiliary data is a small fraction of the mix.
>
> We do not yet have a prescription that can be set without knowledge of the capabilities. We are running sweeps at 50M, 100M, and 200M parameters (using "absolute deviation from data filtering's capability profile" as a metric) to study how consistent optimal settings are across scales, and **we will include concrete guidance in the revised paper.**
>
> **W3: Scaling with number of capabilities**
>
> > Figure 4 is not exactly compelling. I'm not sure how to evaluate whether the technique is solid enough to be used with, say, 30 categories.
>
> We are extending this experiment to 32 categories to better understand the scaling trends. (Going beyond 32 would be nontrivial given the limited number of SimpleStories categories.)
>
> **Q1: Why does $p_\text{cr}$ activate a random auxiliary expert?**
>
> We want the model to be robust to the inclusion or exclusion of any auxiliary expert during inference. Random sampling during training reflects the distribution of configurations the core will see in deployment. When activated via $p_\text{cr}$, the auxiliary expert also learns from core data. Intuitively, it makes it easier for the model to "calibrate" to a configuration if both experts can learn in that configuration. Empirically, intermediate $p_\text{cr}$ gives the best validation loss (Figure 11).
>
> A targeted strategy might improve competence but would likely hurt robustness, since the core would learn to depend on specific expert-context pairings.
>
> **Q2: Overlapping capabilities and imperfect filtering**
>
> > You assume capabilities are nonoverlapping... Also, you assume capabilities are weeded out entirely from the core data set.
>
> We respectfully disagree with the reviewer's assessment. As mentioned in response to W1, our settings contain overlapping capabilities and GR-MoE is compatible with this.
>
> Additionally, to further strengthen understanding of the labeling requirements of GR-MoE, we ran a new experiment comparing GR-MoE to data filtering as the fraction of auxiliary data that is labeled varies (unlabeled auxiliary data remains in the core set). As labeling improves, forget CR falls for both methods. At high labeling rates, DF achieves lower forget while GR-MoE achieves higher retain (1.001 vs 0.976 at full labeling). The full table is in our response to Reviewer vYLf, Q1.
>
> We hope these responses address the reviewer's concerns, and respectfully ask the reviewer to consider raising their score if they find them satisfactory.

---

> > ### Author Rebuttal · Reviewer_5fqV · 2026-04-03
> >
> > I appreciate the additional experimental result presented and the fact that there was an appendix which addressed one of my questions. I am very happy to see the paper accepted, but I feel the "weak accept" rating is more reflective of my view than the "strong accept" so I am leaving my score unchanged.

---

### Official Review · Reviewer_kHwQ · 2026-03-14

**Soundness:** 3
**Presentation:** 3
**Significance:** 2
**Originality:** 2
**Overall Recommendation:** 4
**Confidence:** 2

**Summary:**

This paper proposes a new MoE architecture and training algorithm called GR-MoE. The motivation is that, to control the model's capabilities, the most direct approach is a data-centric one, such as training data filtering or unlearning. However, data-centric approaches, such as data filtering, may not be sufficiently scalable or flexible. GR-MoE is a flexible MoE architecture in which each expert is trained for a specific capability. During inference, GR-MoE can specify which experts to activate to control its capabilities for different users or sessions.

**Compliance With Llm Reviewing Policy:**

Affirmed.

**Final Justification:**

My questions have been addressed. I think this is an interesting paper, with conservative confidence.

**Key Questions For Authors:**

See above.

**Limitations:**

See above.

**Strengths And Weaknesses:**

The paper is very well-written and easy to follow.

I think the methodology is very clean. The main weaknesses with the methodology, in my opinion, are the following:

1. GR-MoE architecture may suffer from load imbalancing when scaled to frontier sizes. Because the routing mechanism dictates that an entire batch of domain-specific data is processed by a single auxiliary expert, the current training setup naturally relies on standard data parallelism. However, scaling to massive models that necessitate expert parallelism would create a severe hardware bottleneck. If a homogeneous batch is routed to just one expert, the compute nodes hosting those parameters would be overwhelmed, while the rest of the cluster sits idle.

2. The methodology's reliance on "safety through unawareness" may be insufficient for highly capable models. GR-MoE assumes that ablating a domain-specific expert effectively removes the parameters responsible for a targeted harmful capability. Yet, as models scale and develop advanced general reasoning skills within the shared core expert, they may be capable of deriving harmful responses, either directly or through in-context learning.

While this method may not be practical at scale due to the above issues (happy to be corrected), I think it's still an interesting idea. I set my score to be 4. Since I do not directly work in this field and I am not familiar with the most recent literature, I set my confidence to be 2.

---

> ### Author Rebuttal · Authors · 2026-03-31
>
> # Response to Reviewer kHwQ
>
> We thank the reviewer for their thoughtful comments. We are glad they found the paper well-written and the methodology clean.
>
> **W1: Load imbalancing at frontier scale**
>
> > GR-MoE architecture may suffer from load imbalancing when scaled to frontier sizes.
>
> The reviewer is right that efficient inference with a large number of auxiliary experts is not fully solved. However, this need not be an issue for efficient deployment of GR-MoE. The core expert is 95% of the MLP dimension and is active on every batch, so most of the compute is in the core regardless of which dataset a batch comes from. The core can be scaled with standard data or tensor parallelism, or could itself be a standard MoE with expert parallelism, while auxiliary experts are treated as small dense adapters similar to LoRA at inference.
>
> **W2: Safety through unawareness**
>
> > As models scale and develop advanced general reasoning skills within the shared core expert, they may be capable of deriving harmful responses, either directly or through in-context learning.
>
> We agree that this is a limitation of the approach. It is also a limitation of data filtering, since a data-filtered model with strong general reasoning might also derive restricted outputs in-context. Forcing an attacker to derive a capability in context rather than access it from weights could provide significant benefits, by increasing the cost of an attack and improving observability for a model API provider. In the paper, we present evidence that GR-MoE approximates data filtering. Insofar as this is true, GR-MoE inherits the aforementioned benefits and drawbacks.
>
> We hope these clarifications address the reviewer's concerns, and respectfully request that the reviewer update their scores if so.

---

> > ### Author Rebuttal · Reviewer_kHwQ · 2026-04-04
> >
> > Thanks for the response. I missed the detail that the core expert is roughly 95% of the model and is always active. If I understand it correctly, running GR-MoE costs the exact same amount of compute as running a standard, dense Transformer. There is no inference speedup or sparse-activation efficiency gain here, and the traditional compute benefit of an MoE is indeed lost here.

---

> > > ### Author Response · Authors · 2026-04-06
> > >
> > > We thank the reviewer for following up.
> > >
> > > The reviewer's understanding of inference costs is correct: GR-MoE inference costs are comparable to a standard dense Transformer. Its purpose is not sparse-activation efficiency but approximating $2^{N-1}$ data-filtered models in one model, in a single training run. The auxiliary experts are best thought of as small, domain-specific adapters (similar to LoRA) that can be selectively ablated to control capabilities at inference time.
> > >
> > > Importantly, **GR-MoE is not intended to replace sparse MoE**. The two are entirely complementary. The large core expert could be implemented as a sparse MoE, recovering the training and inference advantages needed at frontier scale. We see no fundamental obstacle to this combination and have updated the discussion section to highlight this as an opportunity for future research.
> > >
> > > We believe this resolves **W1**: since the auxiliary experts are small and the core is always active, GR-MoE inherits the training and serving profile of its core architecture (whether dense or a standard sparse MoE), so the load-imbalance issue need not arise. If the reviewer's concerns are addressed, we would be grateful if they would consider updating their score.

---

### Decision · Program_Chairs · 2026-04-30

**Decision:**

Accept (spotlight)

**Comment:**

The authors propose a gradient-router MoE architecture for pre-training in order to isolate and control knowledge/AI capability access during inference. The reviewers found the paper well-structured; most of the reviewers agreed that the problem studied is highly relevant. One of the reviewers noted that the method successfully approximates the safety of multiple separately trained models (on different parts of the data) within a single run, thus offering efficiency advantage.

One of the reviewers noted that because the core expert is 95% of the network and is always active, GR-MoE costs the exact same amount of compute as a standard dense transformer during inference, meaning it provides none of the sparse-activation efficiency gains traditionally associated with MoE architectures. While this was not a core concern, it should be clearly noted in the final version of the work.

During the rebuttal, the authors addressed several critical concerns. They clarified the architecture and load balancing; also further clarified that their metric adequately captures the removal of latent abilities, showing robustness to adversarial fine-tuning.

One of the reviewers remained skeptical about the real-world applicability of the problem formulation, rightfully noting that capabilities are rarely separated in practice and that the problem was slightly "formulated to facilitate the solution". This was partially addressed: the authors provided new empirical results showing that GR-MoE remains robust under label corruption and in datasets with higher semantic overlap.